



# Accuracy in starphotometry

Liviu Ivănescu[1], Konstantin Baibakov[1,2], Norman T. O'Neill[1], Jean-Pierre Blanchet[3], and Karl-Heinz Schulz[4]

[1]Centre d'applications et de recherches en télédétection (CARTEL), Université de Sherbrooke, Sherbrooke, QC, Canada
[2]Canadian Space Agency | Agence spatiale canadienne, Saint-Hubert, QC, Canada
[3]Department of Earth and Atmospheric Sciences, Université du Québec à Montréal (UQÀM), Montréal, QC, Canada
[4]Dr. Schulz & Partner GmbH, Falkenberger Str. 36, Buckow, Germany (end of operations as of April 2016)

**Correspondence:** Liviu Ivănescu
(Liviu.Ivanescu@usherbrooke.ca)

**Abstract.** Starphotometry, the nighttime counterpart of sunphotometry, has not yet achieved the commonly sought observational error level of 1%: a spectral optical depth (OD) error level of 0.01. In order to address this issue, we investigate a large variety of systematic (absolute) uncertainty sources. The bright star catalog of extraterrestrial references is noted as a major source of errors with an attendant recommendation that its accuracy, as well as its spectral photometric variability, be significantly

improved. The small Field of View (FOV) employed in starphotometry ensures that starphotometry, unlike sun- or moonphotometry, is only weakly dependent on the intrinsic and artificial OD reduction induced by scattering into the FOV by optically thin clouds. A FOV of 45 arc-seconds was found to be the best tradeoff for minimizing such forward scattering errors concurrently with flux loss through vignetting. The importance of monitoring the sky background and using interpolation techniques to avoid spikes and to compensate for measurement delay was underscored. A set of 20 channels was identified to mitigate con-

tamination errors associated with stellar and terrestrial-atmospheric gas absorptions, as well as aurora and airglow emissions. We also note that observations for starsphotometers similar to our high-Arctic starphotometer should be made at high angular elevations, i.e. at airmasses lower than 5. We noted the significant effects of snow crystal deposition on the starphotometer optics, how pseudo OD increases associated with this type of contamination could be detected and how proactive techniques could be employed to avoid their occurrence in the first place. If all these recommendations are followed, one may aspire to

achieve component errors that are well below 0.01: in the process one may attain a total 0.01 OD target error.

## 1 Introduction

The nocturnal monitoring of semi-transparent atmospheric features, such as particles (aerosols, optically thin clouds) or gases ($O_3$, $H_2O$) can be performed using attenuated starlight, to derive a spectral optical depth (OD). The passive remote sensing method of stellar spectrophotometry (known as starphotometry by the atmospheric remote sensing community) was accord-

ingly introduced in the early 1980s (Alekseeva, 1980; Roddier, 1981). Despite some technological progress, accurate stellar spectrophotometry remains a challenge (Deustua et al., 2013; Bohlin et al., 2014). Its evolution, with an emphasis on problems particular to starphotometry, can be followed in Roscoe et al. (1993); Leiterer et al. (1995, 1998); Herber et al. (2002); Pérez-Ramírez et al. (2008a, b); Baibakov et al. (2009, 2015); Ivănescu (2015). The accuracy of the optical depth (OD) retrieval





remains critical for (2$^{nd}$ spectral-order) particle feature extraction methods, which require sub 0.01 optical depth precision error

(O'Neill et al., 2001). Such a precision error would necessitate sub 0.01 accuracy error. Other technical and data processing challenges remain inasmuch as this relatively rare type of instrument, with only a few operational starphotometers worldwide, is still evolving.

Sunphotometry, and to some extend moonphotometry, are much more mature technologies. The current starphotometers cannot yet, for example, parallel the automated robustness of the CIMEL sunphotometers in the AERONET network (see

Holben et al. (2001) for a discussion of the CIMEL instrument and the AERONET network). One can aspire to benefit from the accomplishments of the solar methodology and improve its nocturnal counterpart. An early and comprehensive analysis of sunphotometer related errors and its data processing procedures was detailed in Shaw (1976), with subsequent contributions by (Forgan, 1994; Dubovik et al., 2000; Mitchell and Forgan, 2003; Cachorro et al., 2004).

OD retrieval, typically in the near-UV to near-IR spectral range, is based on the Beer–Bouguer-Lambert law of atmospheric

attenuation. The detailed heterochromatic (wide spectral band) attenuation law was investigated by King (1952); Rufener (1963, 1986); Young and Irvine (1967). While employing wide spectral bands enhances the S/N (signal to noise ratio) of faint stars, the attenuation law is substantially simplified in the monochromatic approximation. Depending on the acceptable error, the approximation is generally valid for spectral bandwidths narrower than 50 nm (see Golay (1974), pages 47–50). The narrow bands typical of sunphotometry are also employed in starphotometry: however accuracy requirements generally limit

the operational star set to the brightest stars (visual magnitudes less than 3).

Beyond the fact that stellar photometric observations are currently not accurate enough, the lack of information on certain types of errors is even more problematic. Our purpose is to overcome such issues and enhance the starphotometry reliability. A comprehensive initial analysis of stellar photometry errors was detailed in Young (1974). Strategies for retrieving accurate photometric observations in variable optical depth conditions were proposed by Rufener (1964, 1986). Those fundamental

astronomical studies remained largely unreferenced in atmospheric science literature. In the present study we invoke and complement them in order to identify and characterize most sources of systematic uncertainty. We expect that, with the proper approach, optical depth accuracy within 0.01 is achievable. That target aside, the very act of approaching this value, is worthwhile as it will increase the level of trust and reliability in starphotometry. We seek to achieve such a goal by identifying ways to mitigate the most important errors, whether by virtue of instrumental and/or retrieval-algorithm improvement or by improved

observational strategies.

The paper consists of instrumental descriptions and a comprehensive development of OD retrieval methods followed by a detailed discussion of the error sources associated with each key OD retrieval parameter. It concludes with recommendations for achieving the 0.01 OD error goal. Most of the errors we describe are of a general nature, while some are specific to our particular spectrometer-based starphotometers (Ivănescu et al., 2014).

We only focus on accuracy aspects, leaving precision and calibration errors to be addressed in subsequent studies. We also avoid the non-linear complications associated with measurements in the water vapor absorption bands (in the neighbourhood of 940 nm): this subject has already been extensively described in the studies of Galkin and Arkharov (1981), Halthore et al. (1997), Galkin et al. (2010a) and Galkin et al. (2010b).



## 2 Observing conditions

As detailed in Appendix A, the data reported in this paper was acquired using two similar instrument/telescope configurations that were designed and built by Dr. Schulz & Partner GmbH: the identical SPST05 and SPST06 instruments with Intes Micro Alter M703 telescope, and the upgraded SPST09 instrument with Celestron C11 telescope (all being spectrometer-based photometers). In Appendix A FOVs of $57.3''$ and $36.9''$ were inferred, respectively, for the earlier and upgraded instruments. SBIG CCD cameras are employed for star acquisition: their native camera pixels are binned into larger pixels of $3 \times 3$ native

pixels, with an angular resolution of $3''$/bin for the SPST05/M703 and $2''$/bin for the SPST09/C11 instrument. Other technical parameters of the most recent version (SPST09/C11) are listed in Table 1.

The simultaneous measurement of all channels by all three spectrometer based systems renders them particularly appropriate for observing rapidly evolving atmospheric features, such as optically thin clouds. This is important for purposes of coherent spectral analysis where all the channels have to capture the same sky view. Other starphotometer types are filter wheel based

systems that sequentially observe one channel at a time (see, for example Leiterer et al. (1995); Herber et al. (2002); Pérez-Ramírez et al. (2008b)).

The observation sites included a variety of environments: warm, continental environment at the mid-latitude sites of Egbert and Sherbrooke; warm, continental and marine environment at the mid-latitude site of Halifax; warm and dry, tropical high altitude site influenced by frequent Saharan dust events at Izaña; marine environment at the low Arctic site of Barrow and

a cold and dry environment, influenced by the quasi-constant presence of ice crystals at the low altitude, high Arctic site of Eureka. The latter is unique in terms of its extreme environmental conditions and the deployment of a larger telescope (C11). More details about the Eureka instrument and the observing facility (shown in Figure 1), as well as its remote operation are found in Ivănescu et al. (2014). One particular consideration of note in this case is the recurring frost formation on the telescope corrector plate and the quasi-constant deposition of ice crystals on it.

In Figure 2 we show observations, at Eureka and Sherbrooke, of star spot sizes (FWHM $\equiv \omega_s$ for short, quasi-instantaneous exposures and $\omega$ for long time exposures) as a function of the observing airmass. For the development of Figure 2 we employed 5–40 short exposures per recording position. The ($\omega_s$) exposure (integration) times were star dependant: they were varied from 1–30 seconds to avoid detector saturation for a given star. The exposure-to-exposure position change on the CCD of these short-exposure spots (the blue and black dots of Figure 2), is largely influenced by turbulence jitter (Roddier, 1981). To account for

this aspect and fully characterize the turbulence, one artificially creates long-exposure (1–4 minute) spots by adding up the short exposure spots on the CCD. The ($\omega$) spot size of such a synthesized superposition of smaller spots will inevitably be relatively large and will be an average indicator of turbulence. We should note that the standard starphotometry integration times (6 s) are similar to those employed for Figure 2 short exposure times: the reason that we create the long-duration spot size is to adequately characterize the low frequency component of the turbulence. In this sense, an estimate of the true or

total (all-frequency) turbulence requires the artificial generation of the long-exposure spots (a problem that is rather unique to bright-star starphotometry due to detector saturation concerns). Figure C1 and Figure C2 show, respectively, a schematic representation of a short-exposure star spot and two measured short-exposure star spot images acquired by the SBIG high-





**Table 1.** Technical parameters of SPST09

| | |
|---|---|
| Telescope | Schmidt-Cassegrain C11 Celestron, aperture ($D$) 280 mm, focal length ($f$) 2800 mm |
| Measurement range | 399.1 – 1159.3 nm, resolution 0.7 nm |
| Standard channels | 17 channels: 420, 450, 470, 500, 532, 550, 605, 640, 675, 750, 778, 862, 934, 943, 953, 1024, 1040 nm |
| FOV | 36.9″ |
| Wavelength error | ± 2 nm |
| Diffraction method | grating |
| Spectral bandwidth | FWHM ≃ 8.2 nm |
| Detector | CCD sensor S7031 (Hamamatsu) |
| Number of pixels | 1024×58 (1044×64 total), 24.6 $\mu m^2$ |
| Quantum efficiency | 90% peak |
| ADU | 22 $e^-$/cnt |
| Standard exposure | 6 sec |
| Time resolution | < 3 min for OSM, < 6 min TSM |
| Star mag. range | < 3 |
| OD accuracy | 0.003 – 0.011 |
| Guiding system | two SBIG CCD cameras |
| Tracking system | mounts: GTO900, AZA2000, G11 |
| Operating temperature range | down to -80°C (with additional temperature insulation and heating) |
| Interface | RS232 |
| Power supply | 12V (3 A) |
| Instrument weight | 13 kg |
| Telescope weight | 14 kg |

resolution camera (for two of the points on Figure 2). Details on the theoretical and empirical context needed to understand the star spot computations is given in the associated text (Appendix C).

In order to avoid any flux loss, the photometer FOV must be much larger than the FWHM of the short exposure image, whose intensity profile (the star Point Spread Function, or PSF) can be approximated with a Gaussian profile (Racine, 1996). The total FWHM is then quadratically composed of the $\omega$=FWHM of the "seeing" spot (the blurring due only to the air turbulence) and $\omega_d$=FWHM of the Airy diffraction spot (approximated by a Gaussian profile whose FWHM is set equal to the FWHM of the diffraction spot). Optical aberrations, especially coma for this type of telescope, may also play a role. However, tests done at

AiryLab (2012) show that the C11, when correctly collimated, is not subject to optical aberrations that influence the size of


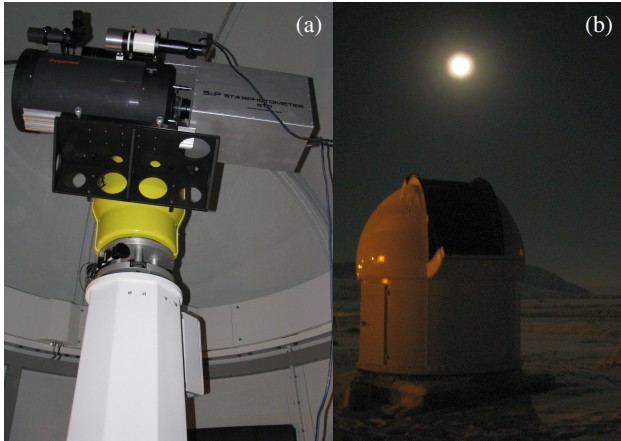

**Figure 1.** The SPST09 starphotometer and C11 Celestron telescope installed on the AZA-2000 mount, inside the Baader dome in Eureka (a). Outside view of the dome in Eureka, during a starphotometer observation (b).

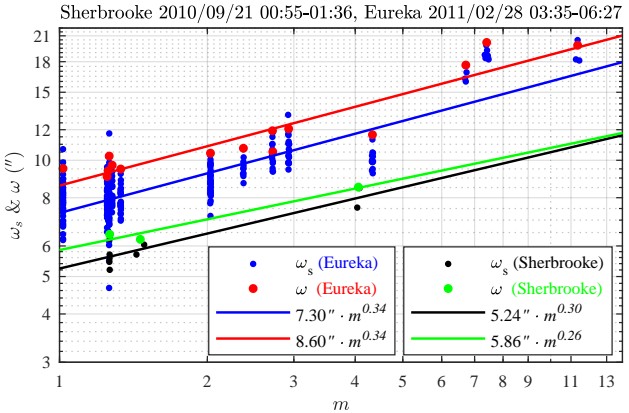

**Figure 2.** Very large star spots measured at the mid-latitude site of Sherbrooke, Québec, Canada and the high-Arctic site of Eureka, Nunavut, Canada (M703 and C11 telescopes, respectively) show a weaker airmass dependence than expected. The symbol $\omega_s$ is associated with short time exposures, while $\omega$ represents long time exposures.

such large star spots as those of Figure 2. The angular size of an Airy spot can be computed as $\omega_d = 1.03 \cdot \lambda/D$, with $\lambda$ being the measurement wavelength. This gives less than $1''$ ($0.49''$ for C11 and $0.75''$ for M703), at $\lambda = 640$ nm (peak of CCD detection). Since these values are 10–20 times smaller than the star spots, the observed FWHM is practically $\omega$. Figure 2 indicates that, for typical atmospheric remote sensing sites (near sea-level, not particularly dry, near heated buildings etc.), the expected seeing
could be $\sim 10$ times larger than what is usual in professional astronomy ($\sim 1''$). Uncontrolled telescope motion in strong winds may also increase the size of the recorded star spots. However, for the observational conditions associated with Figure 2, the surface wind impact was negligible.



The turbulence strength can be assessed through the length parameter $r_0$ (Fried, 1966). If we apply the $\omega$ values of Figure 2 to the expression of Racine (1996) ($\omega = 0.976 \cdot \lambda/r_0$) we obtain $r_0$ in the 5–15 mm range (about the size of the inner scale of turbulence). This means that the turbulence goes beyond the inertial Kolmogorov spectrum, normally producing star spot sizes dependent on $m^{0.6}$ (i.e. $m^{3/5}$) (Roddier, 1981), into the dissipation regime of the von Karman spectrum (Osborn (2010), pages 16–17). This may explain the $\sim 0.3$ exponent of $m$ in Figure 2: such a value corresponds to a von Karman spectrum at high spatial frequencies (see, for example, Figure 2.3, ibid). Also, since $\omega$ of Figure 2 corresponds to an averaged $\lambda = 640$, and since $\omega \sim \lambda^{-1/5}$, then $\omega$ should be $\sim 10\%$ larger at 400 nm, and $\sim 10\%$ smaller at 1000 nm.

With respect to the $\omega \simeq 1''$ values usually experienced at high altitude professional (non-amateur) astronomical sites, it's important to note the dramatically large values associated with the sea-level (10 m altitude) Eureka station of Figure 2. However, the seeing at the 610 m altitude Ridge Lab (CANDAC site, also at Eureka) is, relatively very small (Steinbring et al., 2013) and comparable with the best observing sites. One concludes that most of the turbulence at Eureka is confined in the first few hundred meters above sea level. It is instructive to characterize the vertical structure of the turbulence, notably its effect on the refractive index variation and, consequently, on star blurring (see, for example, Owens (1967) for basics on the refractive index of air). Unfortunately, a precise characterization solely based on radiosonde measurements may not be possible (Roddier, 1981). However, that vertical structure can nevertheless be approximated parametrically. Accordingly, we express the vertical variation of the star spot size due to turbulence as

$$dω = k_c \cdot k_t \cdot dn \cdot dv/v \tag{1}$$

where $dv/v$ is the relative wind shear (whose kinetic turbulent energy is the primary influence on the refractive index variation ($dn$) between the atmospheric layers). This equation is a first order, empirically derived to a convenient expression, whose goal was to arrive at a coarse representation of $\omega$ versus altitude. The constant $k_t \simeq 6$ is an empirical normalizing constant that adjusts the right side of equation (1) so that its integration yields the surface-level $\omega$ values of Figure 2. Employing an ensemble of Eureka, polar winter sounding profiles acquired over a $\sim 6$ week period within the Figure 2 measurement period, we integrated the $d\omega^2$ interpretation of those profiles from the maximum altitude of the radiosonde to a given altitude in order to yield $\omega$ at every altitude (Figure 3). On the median (red) and average (green) curves one can identify major blurring increases: just below 3 km, below 200–400 m (suggesting a quasi-permanent turbulent layer), and again about 10–20 m from the surface. This confirms the very low $\omega$ values at the Ridge Lab, despite the dramatically large seeing at sea-level.

## 3 Observing methodology

A photometric system, from the perspective of the astronomical community, is a system assessing the brightness of an object on a logarithmic scale, normalised to a standard reference (a natural source or a convenient synthetic spectrum).



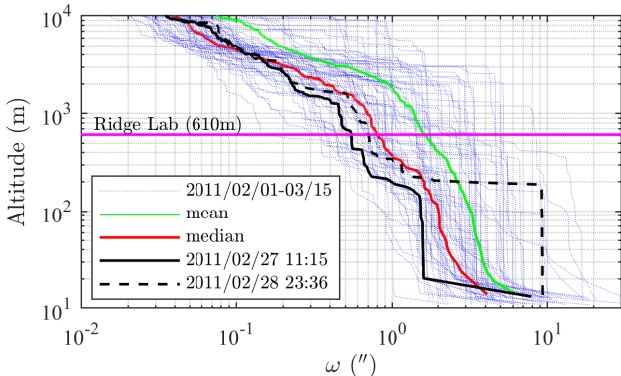

**Figure 3.** Vertical structure of star spot blurring in Eureka derived from the quadrature integration of equation (1). Most of the turbulence is below the Ridge Lab elevation (magenta line). The black curves were derived from the two nearest soundings to the $\omega$ measurements in Figure 2. The $k_t = 6$ (derived for the black curves) was employed for all the other curves.

### 3.1 Catalog photometric system

We denote by $I$ the star irradiance expressed in absolute measurement units. By definition, the apparent magnitude ($M$) of a star is computed from the ratio between $I$ (the observed irradiance) and $I_{0,\text{cref}}$, the unattenuated ("0") irradiance of a "catalog reference" ("cref") source

$$M = -2.5 \log \frac{I}{I_{0,\text{cref}}} = -2.5 \log I + 2.5 \log I_{0,\text{cref}} \tag{2}$$

where "log" is short for "$\log_{10}$". The quantity

$$\text{ZP} = 2.5 \log I_{0,\text{cref}} \tag{3}$$

is usually referred to as the "zero-point" of the photometric system and serves, from a practical standpoint, to identify the photometric system. Star magnitudes are therefore photometric-system dependent. The magnitude of the reference source at any wavelength is, by definition, $M_{0,\text{cref}} = 0$ (i.e. when $I = I_{0,\text{cref}}$). Most of the photometric systems currently employed are based on Vega as primary reference source ("primary standard") (Bessell, 2005).

One can recast equation (2) into its extraterrestrial form

$$M_0 = -2.5 \log \frac{I_0}{I_{0,\text{cref}}} = -2.5 \log I_0 + 2.5 \log I_{0,\text{cref}} \tag{4}$$

The adjective "extraterrestrial" can be also represented by "unattenuated", "extra-atmospheric", "exoatmospheric" or "zero-airmass" in the literature (ground-based measurements are also referred to as "attenuated"). The signature extraterrestrial magnitudes of each star are found in the catalog(s) of various observation campaigns. $M_0$, obtained with the V standard wide-band filter (Johnson and Morgan, 1953), covering most of the visible spectrum, is usually called visual $V$ magnitude. The blue $B$ magnitude is then obtained with their B band filter, etc.



The most accurate exoatmospheric star irradiance catalog in the starphotometry spectral range is Pulkovo's (Alekseeva et al., 1996). This catalog provides near-UV to near-IR $I_0$ spectra[1] for most of the brightest stars ($V < 3$). Its magnitudes (Alekseeva et al., 1994) are simply expressed as

$$M_0 = -2.5 \log I_0 \tag{5}$$

with $I_0$ converted to cgs units of $\left[\mathrm{erg\,s^{-1}\,cm^{-2}\,cm^{-1}}\right]$. From equations (3) and (4), one concludes that ZP = 0 in equation (5).

Therefore, the reference spectrum used to compute the Pulkovo catalog magnitudes is spectrally flat and equal to unity ($I_{0,\mathrm{cref}} = 1\ \mathrm{erg\,s^{-1}\,cm^{-2}\,cm^{-1}}$). Such a unit reference defines a "raw" photometric system (or "raw" magnitudes). Its SI-units value of $0.1\ \mathrm{W\,m^{-2}\,m^{-1}}$ is near the Vega irradiance maximum of $0.0796\ \mathrm{W\,m^{-2}\,m^{-1}}$ at 402.5 nm (when measured at 8.2 nm bandwidth)[2].

### 3.2    Theoretical considerations

The starphotometer measurement principle is based on the Beer–Bouguer-Lambert attenuation law applied to the starlight passing through the Earth's atmosphere (as described, for example, in Liou (2002)). The attenuation, due to the out-scattering and absorption of the incoming light by atmospheric particles and gases, is described by

$$I = I_0 e^{-m\tau} \tag{6}$$

with $\tau$ being the total vertical optical depth, $m$ the stellar airmass, $I$ and $I_0$ the attenuated and unattenuated star irradiances,

respectively. For a plane-parallel atmosphere approximation, $m = 1/\cos\theta$, with $\theta$ being the zenith angle of a given star (the approximation is generally valid for $\theta \lesssim 80°$, or $m \lesssim 6$).

    The law formulated in equation (6) can be more practically converted in a linear form, by expressing it in term of apparent magnitudes ($M$ and $M_0$), as defined in equations (2) and (4). Taking the logarithm of equation (6), one obtains

$$\log \frac{I}{I_0} = -\frac{M - M_0}{2.5} = -m\tau \log e \tag{7}$$

where $e$ is the natural logarithm base. The exponential law then becomes a linear relation in terms of apparent magnitudes

$$M = M_0 + 2.5 \log e \cdot m\tau = M_0 + (m/0.921)\tau \tag{8}$$

This expression, in conditions of approximately constant $\tau$, can be used to retrieve the intercept $M_0$ from a linear regression of $M$ versus $m$. This can be done, for example, by employing a series of irradiance measurements carried out over a clear night with significant changes in $m$ (not always a given in the case of a high Arctic site). Such a procedure is referred to as the

Langley calibration technique, or Langley plot (also described in Liou (2002)).

---

[1]While the Pulkovo catalog irradiances are correctly expressed in SI units of $\left[\mathrm{W\,m^{-2}\,m^{-1}}\right]$ in the VisieR online database (Ochsenbein et al., 2000), their values in the published paper have to be divided by $10^5$ to yield $\left[\mathrm{W\,m^{-2}\,m^{-1}}\right]$.

[2]Where the conversion is as follows: $1\ \mathrm{W\,m^{-2}\,m^{-1}} = 10\ \mathrm{erg\,s^{-1}\,cm^{-2}\,cm^{-1}}$, then $I_{0,\mathrm{cref}} = 1\ \mathrm{erg\,s^{-1}\,cm^{-2}\,cm^{-1}} = 0.1\ \mathrm{W\,m^{-2}\,m^{-1}}$.



## 3.3 Practical considerations

The measured star signal ($F$) is expressed in counts per second (cnt/s). If $F_{0,\mathrm{iref}}$ is the unattenuated "instrument reference" ("iref") signal, defining the instrument photometric system (in cnt/s), the attenuated and unattenuated instrumental magnitudes ($S$ and $S_0$, respectively) can be expressed, in a manner analogous to equations (2) and (4)

$$S = -2.5\log\frac{F}{F_{0,\mathrm{iref}}}, \qquad S_0 = -2.5\log\frac{F_0}{F_{0,\mathrm{iref}}} \tag{9}$$

One can convert $F$ into $I$ with an instrument specific conversion factor

$$c = \frac{I}{F} \tag{10}$$

Applied to the two system references, the ratio becomes

$$c_{\mathrm{ref}} = \frac{I_{0,\mathrm{cref}}}{F_{0,\mathrm{iref}}} \tag{11}$$

This represents a transformation (scaling) factor from the instrument to the catalog reference system. The unitless $c/c_{\mathrm{ref}}$ ratio then incorporates the photometric-system scaling ($c_{\mathrm{ref}}$) as well as the optical and electronic throughput of the instrument ($c$). In terms of magnitudes, we can define the instrument-specific calibration parameter

$$C = -2.5\log\frac{c}{c_{\mathrm{ref}}} = -2.5\log\frac{I}{I_{0,\mathrm{cref}}} + 2.5\log\frac{F}{F_{0,\mathrm{iref}}} \tag{12}$$

Substituting $S$ and $S_0$ from equations (9), as well as $M$ and $M_0$ from equations (2) and (4) into equation (12), yields

$$C = M - S \tag{13}$$

$$C = M_0 - S_0 \tag{14}$$

where the role of $C$ as a conversion factor between the catalog and instrument magnitudes is made readily apparent by the elegant simplicity of this pair of equations.

If the catalog reference is the unattenuated source being observed, then $I_0 = I_{0,\mathrm{cref}}$ and $M_{0,\mathrm{cref}} = 0$ as per equation (4). Accordingly, from equation (14), $C = -S_{0,\mathrm{cref}}$. Alternatively, if the instrumental reference is the unattenuated source being observed, then $F_0 = F_{0,\mathrm{iref}}$ and $S_{0,\mathrm{iref}} = 0$ as per equation (9). Equation (14) then indicates that $C = M_{0,\mathrm{iref}}$ (i.e. the catalog magnitude of the instrument reference source). Equating the $C$ values for those two special cases yields $M_{0,\mathrm{iref}} = -S_{0,\mathrm{cref}}$. In addition, the calibration parameter may be expressed as

$$C = -S_{0,\mathrm{cref}} = 2.5\log F_{0,\mathrm{cref}} = \ln F_{0,\mathrm{cref}}/0.921 \tag{15}$$

with $F_{0,\mathrm{cref}}$ the instrument signal measured when observing the star catalog reference.

In practice, equations (9) are often expressed as

$$S = -2.5\log F = -\ln F/0.921 \tag{16}$$

$$S_0 = -2.5\log F_0 = -\ln F_0/0.921 \tag{17}$$





This either implies that $F$ and $F_0$ are unitless (i.e. measurements are already normalised to the instrument reference), or that
the reference is conveniently chosen as $F_{0,\text{iref}} = 1$ cnt/s (ZP = 0). Such a unit reference, as in the case of the catalog system,
defines a "raw" photometric system (this is the system that is employed for our starphotometers). According to equation (11),
having unit values for both photometric system references, implies that the scaling factor is also unity ($c_{\text{cref}} = 1$). This yields

$$C = -2.5\log c = -2.5\log\frac{I}{F} = -2.5\log\frac{I_0}{F_0} \tag{18}$$

The calibration procedure then reduces to the unattenuated measurement of any source of known irradiance. Equation (18)
may be used in laboratory based calibrations, or in "in-situ" calibrations, by measuring any accurately known star spectra.
This may be done in a Rayleigh atmosphere (i.e. without aerosol or clouds), for which the attenuation can be accurately
estimated (Bucholtz, 1995). Such conditions can generally be approximated at high elevation, calibration sites (supported by
some independent estimate of the small but non negligible aerosol optical depth).

If we define, for simplicity

$x = m/0.921$ $\tag{19}$

equation (8) can be rewritten as

$$M = \tau x + M_0 \tag{20}$$

Substituting $M$ from (13) into (20), yields a Langley calibration equation whose ground-based ($\tau$ dependent) component is
expressed in terms of the instrument signal $S$

$M_0 - S = -\tau x + C$ $\tag{21}$

This expression enables the retrieval of $C$ when $M_0$ is provided by a catalog. However, if an accurate $M_0$ spectrum cannot be
found, then equation (14) can be used to transform equation (21) into a pure instrumentation version

$$S = \tau x + S_0 \tag{22}$$

so that a catalog is no longer required. Instead of finding $C$, one has to employ Langley calibrations to estimate $S_0$ for all
stars that are part of the operational protocol of a given starphotometer. Equation (21), on the other hand, has the advantage
of casting the calibration procedure in terms of an explicit function of a single star-independent constant ($C$). $C$ represents an
intrinsic parameter that remains constant as long as the instrument characteristics do not change.

### 3.4   Measuring methods

#### 3.4.1   One-star method (OSM)

Considering that the main purpose of starphotometer measurements is to retrieve the optical depth ($\tau$), we rearrange equa-
tions (21) and (22) to yield

$$\tau = \frac{S - S_0}{x} = \frac{S - M_0 + C}{x} \tag{23}$$





Restricting measurements to one star speeds up the acquisition process. This is particularly useful in the presence of rapid $\tau$ variations that one observes, for example, during cloud events. However, since equation (23) contains calibration values, any optical or electronic degradation of the instrument will propagate into the $\tau$ estimation.

### 3.4.2 Two-star method (TSM)

The Langley calibration enabled by equation (21) allows the direct retrieval of $\tau$ as the slope of a linear regression between $S$ and $x$. In lieu of such a lengthy procedure (typically requiring hours of measurements acquired over a large range of x) or directly applying the instantaneous OSM equation (23), one can use the measurements of two different stars at two different airmasses in the Two-star method (TSM)[3]. Equation (21) yields

$$\tau = \frac{(S_1 - S_2) - (M_{01} - M_{02})}{x_1 - x_2} \tag{24}$$

where the subscripts "1" and "2" refer to a low star (large airmass) and a high star (small airmass), respectively. In order to minimize OD errors associated with this technique, the airmass difference between the two stars should be large. However, beyond airmass 5, the impact of higher measurement errors may overcome the benefits of a large airmass range (see Young (1974) for an optimization analysis). In practice, the high star is in the range of 1–2 airmasses, while the low star is in the range of 3–5.

The "auto-calibrating" feature of equation (24) (i.e. no need for $C$), is limited in its applicability: there are temporal and spatial restrictions on the variation of $\tau$ between the two observations. It is therefore a method that is more appropriate for the typically weak and slow variations associated with aerosols (as opposed to the typically strong and high frequency variations associated with clouds). There are also restrictions on the optical throughput variation: specifically in terms of any dust, dew, frost or snow deposition on the telescope optics or star vignetting and focusing variations between the two observations. In fact, the TSM can be interpreted as an OSM, with $C$ being determined by regression through only two data points. The TSM-based calibration vector can be obtained from equation (23)

$$C = \frac{(S_1 - M_{01})x_2 - (S_2 - M_{02})x_1}{x_1 - x_2} \tag{25}$$

$$C = \frac{S_1 x_2 - S_2 x_1}{x_1 - x_2} + \Delta M_0 \tag{26}$$

### 3.5 Optical depth accuracy

In reality, we cannot measure the starlight alone, since the measurement always includes a background signal $B$. The latter is mainly due to electronics readout signal and sky brightness. If $R$ is the starphotometer measurement obtained while pointing towards the star, then $B$ can be estimated by a slightly off-axis measurement. In dark sky conditions, $B$ is dominated by the instrument dark current. The desired starphotometer (starlight) signal is estimated as

$$F = R - B \tag{27}$$

---

[3]Also known as the "Δ method" when introduced by Leiterer et al. (1998).





with attendant systematic error components

$$\delta_F = \delta_R + \delta_B \tag{28}$$

For small relative errors $\delta_F/F$, one obtains $\delta_S$ by taking the derivative $|S'|$ of $S$ with respect to $F$ in equation (16)

$$\delta_S = |S'|\delta_F = 1.0857\frac{\delta_F}{F} \tag{29}$$

If the only errors are in $S$, then equation (23) yields

$$\delta_\tau = \frac{\delta_S}{x} = \frac{1}{m}\frac{\delta_F}{F} \tag{30}$$

However, the optical depth accuracy is subject not only to errors in the observational parameter ($S$), but also to all the other
physical parameters ($M_0$, $C$, $x$) involved in the starphotometry retrieval. All the contributions to the line-of-site observation
error can be explicitly listed by differentiating equation (23)

$$\delta_\epsilon \equiv x\delta_\tau = -\delta_{M_0} - \delta_x\tau + \delta_S + \delta_C \tag{31}$$

The other components of the observation error that represent magnitudes ($M_0$ and $C$, as per equations (5) and (18)) can, in a
similar fashion to equation (29), be expressed as

$$\delta_{M_0} = 1.0857\frac{\delta_{I_0}}{I_0}, \qquad \delta_C = 1.0857\frac{\delta_c}{c} \tag{32}$$

A comprehensive description of starphotometry related errors can be found in Young (1974) and Carlund et al. (2003). In the
following sections we continue this work by quantifying the accuracy of each individual parameter of equation (31) ($M_0$, $x$, $S$
& $C$).

## 4    Spectrophotometric catalog ($M_0$) accuracy

In order to move from a star-dependent $S_0$ calibration, which is currently the standard (Rufener, 1986; Pérez-Ramírez et al.,
2011), to the more convenient star-independent calibration in terms of $C$, one has to ensure that the exoatmospheric magnitudes
$M_0$ are sufficiently accurate. The star dataset that we employed (Appendix B) was limited to stars with a maximum of 0.01
magnitude variation in the observations used to generate their Pulkovo catalog entry (that dataset was employed as the default
catalog by the manufacturer of our instruments). They are mostly main-sequence stars (of luminosity class V) (Kippenhahn
et al., 2012) at the most stable period of their life-cycle (five are of luminosity class II–III). Five are of "early-type" spectral
class B stars (i.e. B0–B3), one of "late-type" class A (i.e. A7–A9) and one of class F. They are all characterized by weaker
absorption lines and cleaner continuum (Silva and Cornell, 1992). However, the "early-type" B stars may also experience non-
negligible (0.01 magnitudes) photometric variability (Eyer and Grenon, 1997). Beyond their intrinsic photometric stability,
the $M_0$ accuracy remains a concern. Alekseeva et al. (1996) stated that: "to preserve the uniform absolute system for all our
seasonal catalogues, we always used the same energy distribution of Vega based on the absolute calibrations by Oke and





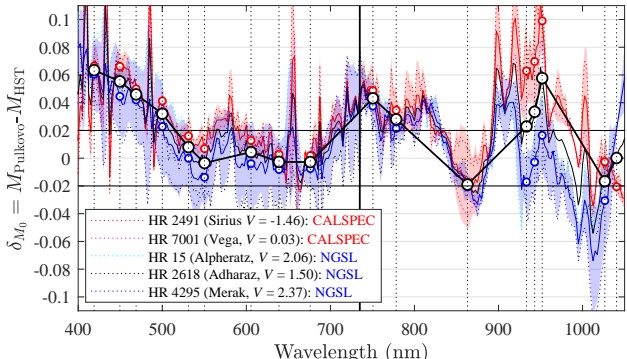

**Figure 4.** Spectrophotometric bias ($\delta M_0$) of the Pulkovo catalog with respect to two different HST catalogs (CALSPEC and NGSL). Open circles represent our standard starphotometer channels, solid colored lines are $\delta M_0$ averages for each HST catalog, while the colored shading represents the corresponding standard deviations. For each spectrum point, the two colored curves and their shading represent sampling populations of 2 points (stars) for the red CALSPEC catalog and 3 points (stars) for the blue NGSL catalog: our objective here was to obtain an estimate of $\delta M_0$ statistics assuming $\delta M_0$ values were roughly independent of the $M_0$ values of individual stars.

Schild (1970) and Kharitonov et al. (1978)". In other words, Vega data was calibrated to the accuracy level achievable about 50 years ago. In addition, Knyazeva and Kharitonov (1990) specified that their (Kharitonov et al. (1978)) calibration values were actually subject to systematic errors that could be as large as 10%. In spite of the shorcomings of the Pulkovo catalog, it remains the most accurate catalog in terms of representing the entire bright star dataset of Appendix B. By comparison, the Hubble Space Telescope (HST) dataset includes only a few of those stars. To better understand the impact of the Pulkovo catalog shortcomings, we compared its absolute irradiances with those measured by the HST. This higher accuracy dataset only contains a few bright stars: Vega (HR7001) and Sirius (HR2491) from the CALSPEC Calibration database (Bohlin et al., 2014), and HR15, HR2618 and HR4295 from the STIS New Generation Stellar Library (NGSL) (Bohlin et al., 2001). Inasmuch as HST measurements are performed with a more recent technology, are not subject to atmospheric effects and have absolute errors below 1% (Bohlin, 2014), we considered them to be the reference. The corresponding magnitude differences between the Pulkovo and HST spectra, computed in terms of the Pulkovo photometric system, are presented in Figure 4. Within a context of the potential impact of atmospheric errors, it is remarkable for a catalog derived from ground-based measurements, that more than half of the standard starphotometer channels (open circles) are characterized by errors of less than 2% or equivalently $\delta M_0 < 0.02$ (equation (32)). Based on the average difference of Figure 4, one nevertheless concludes that the Pulkovo catalog is characterized by a bias that is particularly large in the near-UV and in the 900-1000 nm range. These biases may, in part, be attributable to uncertainties related to the stronger aerosol scattering effects in the UV and to water vapour effects in the near-infrared (NIR). The average bias found in Figure 4 could then be used to correct the Pulkovo catalog. However, a bias will not actually affect the optical depth measurements. For example, in the TSM mode, such a bias is canceled out in the $M_0$ magnitude difference of equation (24). Even in the OSM mode of equation (23), the bias will actually propagate into $C$ during the calibration process. This bias transfer is attributable to the fact that a bias will only affect the intercept of the




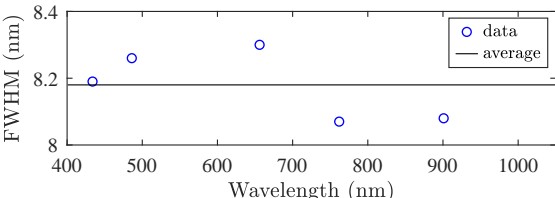

**Figure 5.** SPST09/C11 bandwidth measured with Vega.

Langley plot, not its slope, as expressed by equation (21). The $\delta M_0$ standard deviation of Figure 4 ($\sim 0.02$), can, on its own merits, be compared with the accuracy of 0.015–0.02 claimed for the Pulkovo catalog (Alekseeva et al., 1996), although these values increase in the UV and water vapour channels. One should also note that for its primary reference stars, such as Vega and Sirius, the 0.02 Pulkovo catalog upper limit of error is halved. Such error levels will impact information extraction from optical depth spectra, especially as the required accuracy for aerosol retrievals sensitive to higher orders of the AOD spectrum is $\sim 0.01$ (O'Neill et al., 2001).

Figure 5 shows the quasi-constant $8.2$ nm bandwidth measured by observing Vega with the SPST09/C11 system. Those FWHM estimates are line broadening measures of the strong hydrogen Balmer series ($H_\alpha = 656.3$ nm, $H_\beta = 486.1$ nm, $H_\gamma = 434.1$ nm, but not $H_\delta = 410.2$ nm). These are absorption lines in the star's own atmosphere and are accordingly intrinsic to the exoatmospheric stellar spectra. We also employed the telluric (i.e. Earth's atmosphere) $O_2$ line at 762 nm and another near-infrared line specific to Vega. The observations used for the Pulkovo catalog were, in contrast, made at 5 nm bandwidth over the 310–735 nm range and at 10 nm over the 735–1105 nm range (at 2.5 nm nominal resolution). For bandwidth consistency over the entire 310–1105 nm range, Alekseeva et al. (1996) re-processed the 5 nm measurements to synthesize a unique 10 nm bandwidth. Currently, we only use the 10 nm bandwidth version over the entire 310–1105 nm range. However, as noted in Young (1992), a bandwidth mismatch between the catalog and the instrument (i.e. 10 and 8.2 nm, respectively in our case), may have an impact on the optical depth error and merits investigation. In order to asses the impact of the bandwidth mismatch, we compared the magnitude errors when using $M_{5.0}$ and $M_{10}$, associated with the 5 nm and 10 nm bandwidths, instead of the actual magnitude $M_{8.2}$ at 8.2 nm bandwidth. We also assessed how a simple magnitude calculation $(M_{5.0}+2M_{10})/3$ compares with the actual 8.2 nm bandwidth, in order to improve the actual 10 nm bandwidth catalog. We synthesised star magnitudes for those three different bandwidths by applying Gaussian bandpass filters to the HST data (originally at 1 Å resolution). This is, in fact, a convolution operation that effectively blurs the stellar absorption lines.

In Figure 6 we compare the magnitudes computed for the three bandwidths, for a star of spectral class A0 (Vega). Figure 6a shows a spectral zoom about the 420 nm starphotometer channel. The increased broadening with increasing FWHM about the $H_\gamma$ and $H_\delta$ Balmer lines demonstrates the blurring effect of the different bandwidths. The graph also shows that one may actually limit the blurring impact by optimizing the spectral location of a given channel. Moving the 420 nm channel to 423 nm will, for example, significantly reduce that impact. Figure 6b shows the contamination due to different blurring levels for the entire spectrum (contamination expressed in terms of $\delta M_0$, which from equation (31) is, in the absence of other errors,





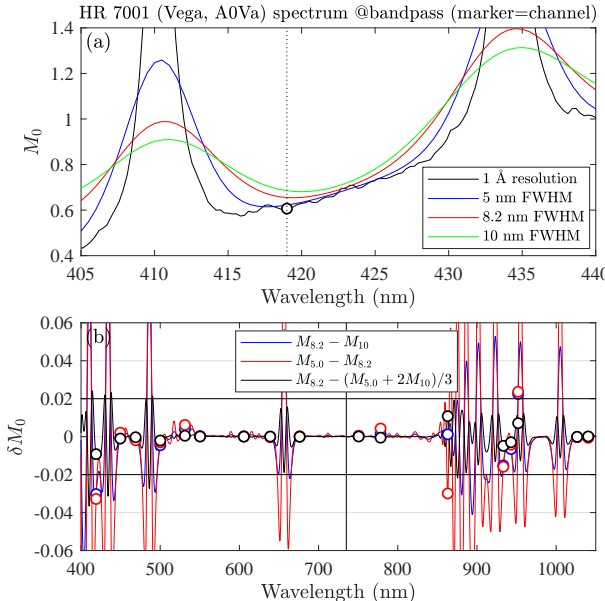

**Figure 6.** Bandwidth mismatch error for a star of spectral class A (Vega, HR7001). Open circles are the nominal starphotometer channels.

equivalent to $x\delta_\tau$. The spiky, high frequency nature of the $\delta M_0$ spectra demonstrates that, while most of the starphotometer channels have negligible ($< 0.01$) errors, channels in the blue and the near-IR are significantly affected. The black curve

"$\delta M_0 = M_{8.2} - (M_{5.0} + 2M_{10})/3$" demonstrates that one may approximate a spectral convolution using a simple average of twice the upper and once the lower bound magnitudes.

The same exercise carried out for a star of early-type spectral class B (Adharaz) underscores the fact that the $H$ Balmer lines are much weaker (Figure 7a). One expects a similar behaviour for our "late-type" A and F class stars. Consequently, the blurring contamination over the entire spectrum (Figure 7b) is, for the case that concerns us the most ($M_{8.2} - M_{10}$) largely

less than 0.01, except for the 958 nm channel that is too close to the 954.6 nm $H$ Paschen absorption line. Inasmuch as all our operational stars are of class A and B (except for one F class star), this analysis is representative. Since the bandwidth mismatch error is a bias that differs for the two star classes of Figures 6 and 7, it may be minimised by distinct photometric calibrations for each star class. However, this may be of limited applicability since the local sky does not present a sufficient array of photometrically stable stars of early-type B, late-type A and F spectral classes.

Up until this point, we have presumed a stable spectral calibration of the instrument. In Figure 8 we show SPST09 spectral drift over almost three years (including four winter seasons) for four stellar-atmosphere absorption lines (hydrogen Balmer series) and two Earth-atmosphere absorption lines (Fraunhofer A of $O_2$ and a NIR $H_2O$ line). Since the stellar lines may shift naturally (for example in the case of pulsating or spectroscopic binary stars), the Earth-atmosphere lines enable both the NIR characterization of the spectrum and a means to approximately monitor the drift at shorter wavelengths. The general shape of

the Figure 8 temporal (Vega) curves was also observed for other stars. The result indicates the a maximum spectral amplitude





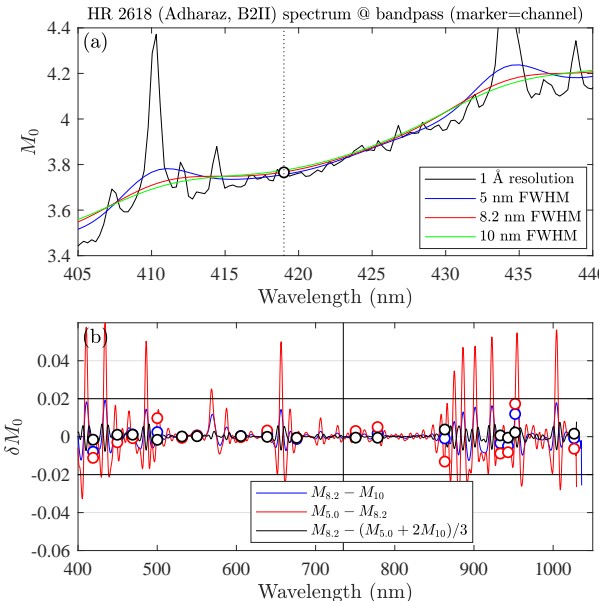

**Figure 7.** Bandwidth mismatch error for a star of (early-type) spectral class B (Adharaz, HR2618). Open circles are nominal starphotometer channels.

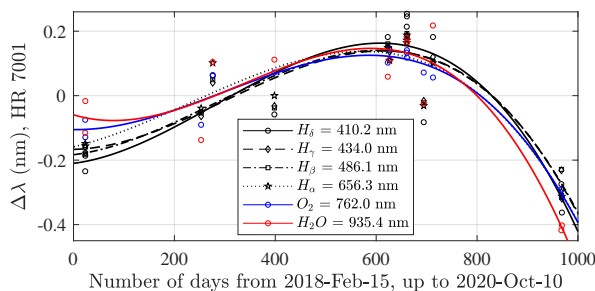

**Figure 8.** SPST09 spectral drift over several seasons for the stellar hydrogen absorption lines of the Balmer series and the atmospheric $O_2$ and $H_2O$ lines. The curves are $3^{rd}$ order polynomial fits.

of 0.5 nm from one year to the next. Such a spectrally variable drift is particularly harmful inasmuch as it will likely influence the spectral shape of the photometric calibration vector (i.e. $C$). A second consequence is that the channels may be subject to additional stellar absorption line contamination if the drift moves them closer to those lines.

A third broadband consequence of the spectral drift results from the stellar-magnitude spectra being generally characterized
by a significant positive spectral slope, over the 400-1100 nm range, for both, A and B class stars, (c.f. Figure 9a and Figure 10a, respectively). This shift in wavelength transforms into a spectral incoherency between the catalogued $M_0$ values and the measured signal. The $M_0$ bias, corresponding to the positive-slope stellar spectrum of Figure 9a for $\pm 0.5$ and $\pm 1$ nm shifts is





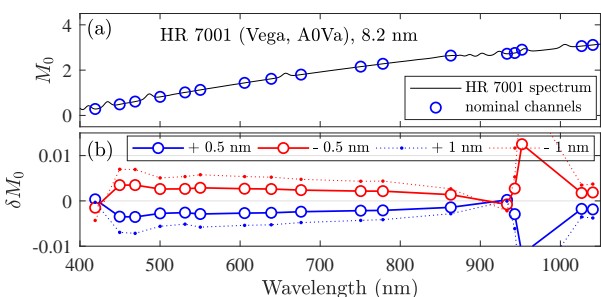

**Figure 9.** Bandwidth mismatch error for an A class star (Vega, HR 7001), as a consequence of a spectrum shift.

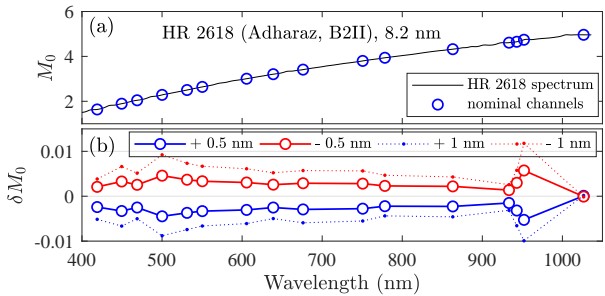

**Figure 10.** Bandwidth mismatch error for an (early-type) B class star (Adharaz, HR 2618), as a consequence of a spectrum shift.

illustrated in Figure 9b. These results indicate that the maintenance of photometric bias values below 0.01 magnitudes requires a spectral calibration within 1 nm (excluding the case of strong water vapour absorption in the near-infrared). The same exercise
is presented in Figure 10 for the early-type B star. While there are individual channel differences with respect to the class A star, the broad $\delta M_0$ results are similar because the $M_0$ slopes are similar.

As long we employ the same class (similar spectral signatures) for both high and low TSM stars, any spectral drift is mitigated in real-time (i.e. similar $\delta M_0$ trends produce common biases and thus the type of bias mitigation discussed in the case of Figure 4 will prevail). While the bias in the OSM case will be initially absorbed into the calibration constant, any
additional drift will progressively propagate into post-calibration $\delta_\tau$ error. Based on the analysis of Figure 8 an annual spectral calibration (preferably at the beginning of the observing season), will likely ensure that the spectral drift be constrained to values $\lesssim 0.5$ nm with negligible effect on the measurement accuracy. Our experience indicates that the six absorption lines employed in the development of Figure 8 are sufficient to adequately characterize the spectral shift of all the starphotometer channels. The radial velocity (stellar center of mass moving away or towards an Earth-bound observer) of our Eureka stars, as
retrieved from (Wenger et al., 2000), lead to 0.15 nm maximum Doppler spectrum shift at 1000 nm, and 0.06 nm at 400 nm, among our stars. Therefore, this effect can be neglected during spectral calibration.

An $M_0$ catalog whose bandwidths match those of the instrument is preferred in order to avoid bandwidth mismatch errors. One natural approach would be to generate a $S_0$ catalog by calibrating the starphotometer at a high altitude site. A single





calibrating site may not, however, yield a sufficient number and class diversity of $S_0$ values (i.e. a sufficiently comprehensive
catalog of stars) to satisfy the starphotometry requirements of a given operational starphotometer site. For spectrometer based
starphotometers, it is necessary to retrieve $S_0$ at all available spectrometer channels (not just the nominal operational channels)
since the spectral drift calculations need to be done at the highest resolutions. This $S_0$ catalog can then be transformed into
a corresponding $M_0$ catalog by first resampling HST $M_0$ values of a selected reference (Vega or Sirius) to the spectrometer
resolution and then employing equation (14) to compute $C$. With that HST-derived value of $C$ in hand, the same equation
can be rearranged to yield $M_0 = C + S_0$ values for all the other stars. Accurate $C$ values and spectral calibration may also
be obtained in the laboratory with the help of a halogen calibration lamp (Paraskeva et al., 2013), or by doing simultaneous
measurements on site with a collocated calibrated instrument.

The alternative to an instrument specific catalog is to use a general purpose high resolution spectrophotometric catalog, from
which one can synthesize magnitudes at any bandwidth (as we did with the HST spectra). Given the maximum bandwidth
mismatch errors found in the Pulkovo catalog ($\sim 0.04$ in Figure 6b for standard channels, at $8.2$ nm bandwidth), we estimate
that a catalog with about 1 nm bandwidth, i.e. about a factor 10 less, would be enough to limit the errors to $< 0.01$. We note
that the generally sub 0.01 mismatch errors estimated for a 1 nm spectrum shift (Figures 9b and 10b) are not inconsistent with
this affirmation. In general a higher resolution catalog such as the HST catalog, with its 1 Å, resolution would be preferred. It
is however surprising that there are no existing high resolution, near-UV to near-IR, spectrophotometric catalogs that achieve
1% accuracy (Kent et al., 2009) for the bright ($V < 3$) stars. The stars observed by professional astronomers are usually
much fainter ($V > 6$) in order to avoid saturating the detectors. This may explain the lack of interest from the astronomical
community in improving the absolute spectrophotometry of bright stars. An effort to address this situation was pursued by
Le Borgne et al. (2003), with their release of the STELIB catalog. However, we identified large biases in the blue/UV part of
the STELIB spectra (Figure 11) in comparison with the HST NGST catalog. The fact that the Pulkovo catalog also has the
largest bias in that range (Figure 4), suggests a recurring issue for catalogs generated from ground-based observations (perhaps
due to the higher optical depth in the blue, and the deficient compensation for aerosol contributions), and accordingly, that
an accurate catalog must be of extraterrestrial origin. Most of the ground-based measurements are focused on achieving 1%
accuracy using broad-band photometry (Stubbs and Tonry, 2006). It is noteworthy however that Zhao et al. (2012) reported
a new spectrophotometric (high spectral resolution) catalog (including our entire bright star dataset) derived from LAMOST[4]
measurements that approached the same 1% accuracy. The spectral resolution and bandwidth of this catalog are variable, but
always sub-nm. The spectral range extends over most of our spectrum, but unfortunately not beyond 900 nm. A novel future
approach for improving the ground-based catalogs would be to employ an accurately calibrated satellite light source in order
to perform stellar differential photometry (Albert, 2012; Peretz et al., 2019).

As an alternative to satellite-based catalogs, the recent ACCESS rocket project (Kaiser and Access Team, 2016) was also a
promising initiative, given their mandate to perform high spectral resolution photometry near the top of the atmosphere. Un-
fortunately their list of $V < 3$ bright stars is limited to Sirius and Vega. Another recent initiative is the NIRS STARS campaign
(Zimmer et al., 2016), whose mandate is to produce a bright star spectrophotometric catalog using lidar measurements to back

[4]Large Sky Area Multi-Object Fiber Spectroscopic Telescope

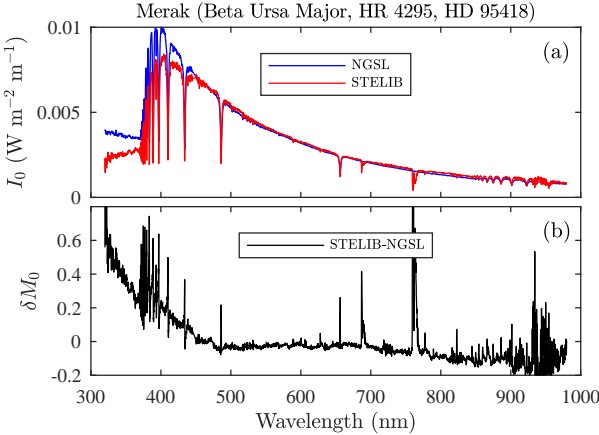

**Figure 11.** Spectrophotometric comparison of STELIB catalog with respect to the HST-NGSL (a). Important bias shows up in the UV and a much weaker one in the IR (b).

out the atmospheric contribution. However, once again, the brightest stars ($V < 3$) are largely excluded from consideration. The most promising option is the use of GOMOS satellite-based star observations (Kyrölä et al., 2004). This sensor employs
high resolution (1.2 and 0.2 nm, depending on spectral bands) limb starphotometry to retrieve ozone and other atmospheric components from space. Its off limb measurements, performed before each limb scan, can be used to build an exoatmospheric spectrophotometric catalog (Ivănescu et al. (2017)). Unfortunately, GOMOS' spectral ranges of 250–675 nm, 756–773 nm and 926–952 nm don't cover our entire 400–1100 nm spectrum. They do, however, cover the problematic spectral ranges experienced in ground-based measurements (the UV/blue and across the $O_2$ and the $H_2O$ absorption bands). The missing portions of
the starphotometer spectra can be filled in by fitting the STELIB, LAMOST spectra, synthetic spectra (Rauch et al., 2013) or averaged star-type spectra (Pickles, 1998) to the GOMOS measurements.

Beyond that, the broadband photometric stability of bright stars remains an open question (as emphasized in Appendix B) and has to be investigated. A uniform photometric variation over the entire observed spectrum may, however, be less critical than a non-uniform one. In an example of the latter case, star temperature variations would lead to spectral distortions with
potential impacts on aerosol retrievals. The analysis of the GOMOS measurements should enable a characterization of spectral variability. If however, an insufficient number of stable $V < 3$ stars are found (i.e. having differential $M_0$ variations of $< 0.01$ between channels), the use of fainter stars may be necessary: this would require the deployment of a larger starphotometer telescope. We will continue, in the short term, to employ the Pulkovo catalog spectra for the operational $M_0$ values of our star dataset. However we will use a synthesised 8.2 nm version, over the available spectral range, to mach our starphotometer
bandwidths.





## 5 Airmass ($x$) accuracy

Systematic errors in the calculation of the airmass $m$ (or alternatively $x$) can be significant (see, for example, Rapp-Arrarás and Domingo-Santos (2011) for a review of analytical airmass formulae). The following operational equation characterizes $m$, in a spherically homogeneous, dry-air atmosphere with an accuracy of better than 1% at $m = 10$ (Hardie, 1962):

$$m = \sec z - 0.0018167(\sec z - 1) - 0.002875(\sec z - 1)^2 - 0.0008083(\sec z - 1)^3 \tag{33}$$

where z is the apparent zenith angle (the zenith angle of the refraction-dependent telescope line of sight). This expression only departs significantly from the plane parallel expression of $m = \sec z$ at values of $m > 5$. If the target star position is computed using astronomical data rather than a measured instrumental mount position then it is more appropriate to use the true zenith angle ($z_t$) formula of Young (1994). The computation of $z_t$ can be effected using star coordinates, site location and time. It ensures an associated maximum 0.0037 airmass error at the horizon (with respect to calculations made on a standard mid-latitude atmospheric model).

One should note that the airmass depends slightly on the vertical structure of the atmosphere (Stone, 1996; Nijegorodov and Luhanga, 1996): an effect which is particularly distinctive in a polar environment. The relative errors due to such environmental variations are however below 0.2% up to $z \simeq 82°$ ($m \simeq 7$), and below 1% at $z \simeq 87°$ ($m \simeq 15$) (Tomasi and Petkov, 2014). Differences in airmass associated with different atmospheric constituents (Tomasi et al. (1998) and Gueymard (2001)), have negligible impact on the observation accuracy of starphotometry.

In spite of the generally high accuracy associated with airmass expressions, the airmass error can be significant if the recorded time stamps are inaccurate. Stars targeted by our starphotometers are recentered between several (3–5) consecutive exposures: a process that is of variable duration (usually 20–40 s). The airmass associated with the mean of all the measurement times (the one reported) may differ from the airmass associated with the mean observing time (the weighted mean time where the weights are exposure duration times). A $\delta_x$ error in $x$, induced by a $\delta_t$ time error, is equivalent to a measurement error $\delta_\epsilon \equiv \delta_x \tau$ (equation (31)). Figure 12 shows the variation of $\delta_\epsilon$ with altitude (for hypothetical observation sites at different elevations), for a $\delta_t = +30$ s case (i.e. time overestimation leading to $\delta_x > 0$ for a descending star), at $\lambda = 400$ nm, and for three different airmasses in a Rayleigh atmosphere (the condition of molecular scattering domination; see Bucholtz (1995) for the optical parameterization of a Rayleigh atmosphere).

The variation of $\delta_\epsilon$ with $x$ is shown in Figure 13a for observations at 10 m (Eureka elevation) and 2360 m (Izaña observatory elevation). The real $x$ variation at Eureka is weak (near the poles, stars carve out sky tracks that vary little in zenith angle). The $\delta_\epsilon$ variation, for unrestricted variation in $x$ (up to $x = 7$), will be comparable for both sites. Figure 13b shows the corresponding $\delta_\tau$ error for Izaña (solid blue line) growing linearly with $x$, and a dominating $x^2$ dependency demonstrated by the saturation of the $\delta_\epsilon(x)/x^2$ curve (dashed blue line). For this simulated $\delta_t = 30$ s case, $\delta_\tau < 0.01$ even at large $x$. However, the computer time may typically drift about 1 min per year (Marouani and Dagenais, 2008): a scenario where $\delta_\tau$ would be significant. The computer time thus has to be corrected weekly, if not daily (using, for example, a GPS time server).





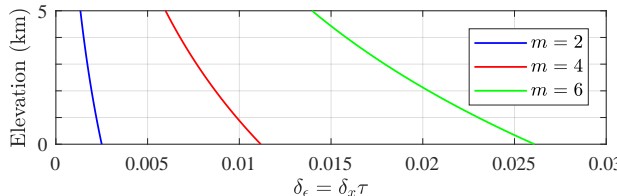

**Figure 12.** Assessment of stellar magnitude errors associated with airmass miscalculation errors due to a time delay error ($\delta_t$) of 30 s in a Rayleigh scattering atmosphere and as a function of the hypothetical elevation of a starphotometer site.

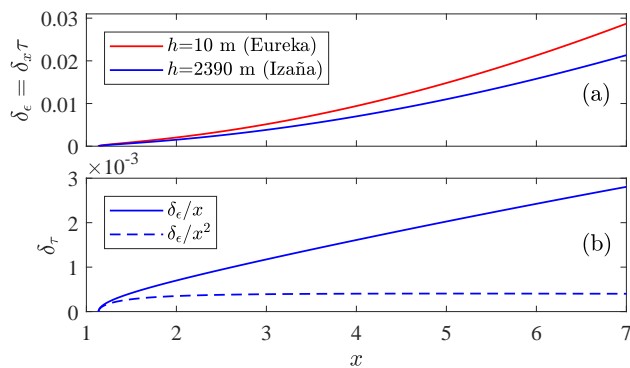

**Figure 13.** (a) Assessment of accuracy associated with airmass miscalculation errors for a descending star (same conditions as Figure 12). (b) $\delta_\tau$ in the case of Izaña site (where $\delta_\tau = \tau\delta_x/x = \delta_\epsilon/x$), while $\delta_\epsilon/x^2$ shows $\delta_x$ dependency in $x^2$.

## 6 Observation ($S$) accuracy

### 6.1 Heterochromaticity

Wide-band optical depth calculations using starlight as the extinction source were first described in Rufener (1964) (in French). A comprehensive description by Golay (1974) (pages 47–50) affirms that non-linear, wide-band radiation detection effects are negligible in terms of $S$ estimation for spectral bandwidths narrower than 50 nm. The error associated with this non-linear component is about the squared ratio between the bandwidth and the central wavelength (i.e. $(\Delta\lambda/\lambda)^2$, Rufener (1986)). A bandwidth of less than 40 nm is then sufficiently small to achieve optical depth errors $< 0.01$ at 400 nm. These optical

depth (heterochromaticity) errors should be well below the negligible value of 0.001 for our sub 10 nm starphotometer-channel bandwidths.

### 6.2 Lognormal fluctuations

The optical depth retrieval, as expressed by equation (23) or (24), is based on computing the instrumental magnitudes $S$ through the logarithm of the measured star signal $F$. However, before doing so, one performs an arithmetic mean $\overline{F}$ over several





consecutive exposures. Since $F$ is subject to log-normal fluctuations induced primarily by scintillation effects (Roddier, 1981), one should characterize its probability distribution in terms of its geometric mean $\overline{\log F}$ and its geometric standard deviation $\sigma_{\log F}$. The corresponding bias, called "misuse of least-squares" by Young (1974), is given by

$$\delta_{\log F} = \log \overline{F} - \overline{\log F} = \sigma_{\log F}^2 / 2 \tag{34}$$

(a classical relationship between the geometric and arithmetic means). From equation (16) and the general definition of a
standard deviation, $\delta_S = 2.5 \delta_{\log F}$ and, similarly, $\sigma_S = 2.5 \sigma_{\log F}$. The bias then becomes

$$\delta_S = \sigma_S^2 / 5 \tag{35}$$

Since a single OD measurement is effectively the arithmetic mean of 3–5 measurements, then only observation fluctuations with $\sigma_S > 0.22$, which we basically never experienced (even at large airmasses), would lead to $\delta_S > 0.01$. One can conclude from equation (30) that $x\delta_\tau < 0.01$ and thus that this issue is negligible in starphotometry.

**6.3    Forward scattering**

Forward scattering into the photometer FOV by atmospheric particulates increases the magnitude of $S$ and thereby induces an underestimate of the optical depth. This "forward scattering error" can be estimated with the single scattering expression

$$\frac{\delta_\tau}{\tau} = \overline{\omega} \cdot P\Delta\Omega \tag{36}$$

where $P\Delta\Omega$ is the integral of the normalised scattering phase function $P$ over the angle $\Omega = \text{FOV}/2$ (Shiobara et al., 1994).
Figure 14 shows a variety of forward scattering error calculations obtained using equation (36) at a wavelength of 400 nm. The red curve represents a typical biomass burning aerosol example (Qie et al., 2017), based on $P$ given by the widely used Henyey–Greenstein (HG) phase function (Zhao et al., 2018). It underscores its negligible forward scattering error on any practical FOV size.

For ice-crystals, $\overline{\omega}$ is practically unity. Two ice-crystal effective diameters were employed: 10 $\mu$m (non-precipitating clouds,
magenta curves) and 120 $\mu$m (precipitating clouds, blue curves). Three crystal habit models were employed to represent the variation of the bulk phase function with crystal habit (from the computations of Baum et al. (2014)): severely roughened aggregates of solid columns (ASC, solid curves, typical in the high Arctic), severely roughened solid columns (Col, dashed curves) and general habit mixture (GHM, dotted curves). Several relevant instruments are represented by vertical black lines in Figure 14: SPST09/C11 (solid), SPST05/M703 (dashed) and the Cimel Sun/Moon photometers with a 1.2° FOV (dash-dotted).
The computations of Figure 14 assume that the contaminating particles (those that induce the FOV scattering effect) are also the particles that one seeks to detect. Those $\delta_\tau / \tau$ computations still apply when the contaminating particles differ from the particles to be detected as long as the FOV effect of the contaminating particles dominates the FOV effect of the latter (for example the FOV effect could be dominated by small OD ice clouds while one seeks to detect fine mode aerosols of significantly larger OD). Optical depth measurements using CIMEL-like instruments in the presence of clouds with $\delta_\tau / \tau$ between 0.15 and
0.5 (the intersection of the dash-dotted CIMEL line with ice cloud curves in Figure 14), means that the $\delta_\tau < 0.01$ requirement





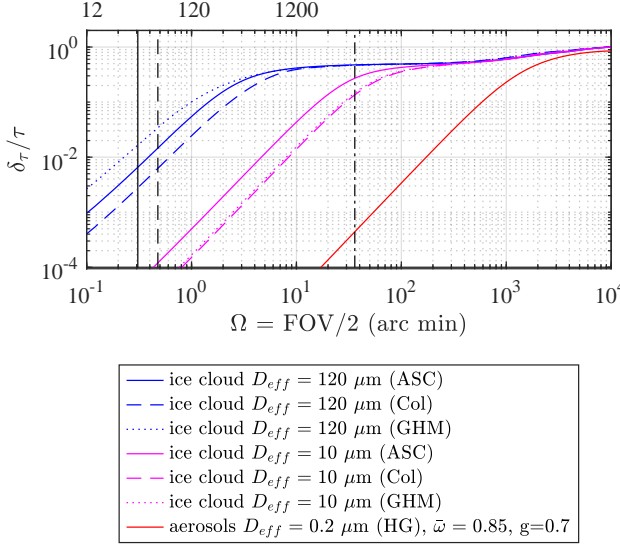

**Figure 14.** The relative forward scattering error for typical aerosols and ice-clouds, as a function of the half field of view. The vertical black lines correspond to SPST09/C11 (solid line), SPST05/M703 (dashed line) and Cimel sun/moon photometers (dash-dotted line). The acronyms between accolades specify the phase function model.

can only be fulfilled for cloud $\tau$ less than 0.015 or 0.05, respectively. If the clouds are thicker than that (which is generally the case), cloud screening is required to ensure accurate AOD. In the case of our starphotometers, those errors are negligible in the presence of non precipitating ice clouds ($D_{eff} = 10$ μm in Figure 14). Even in the case of precipitating clouds ($D_{eff} = 120$ μm in Figure 14), the SPST09/C11 instrument, for which $\delta_\tau/\tau \sim 0.01$, still provides the required accuracy as long $\tau < 1$.

## 6.4 Night sky background

Airglow, and potentially aurora, can be important contributors to the night sky background (see Chattopadhyay and Midya (2006) on the importance of airglow). Their high frequency temporal and spatial variability (Dempsey et al., 2005; Nyassor et al., 2018) complicates their elimination in a background subtraction process. This can lead to significant optical depth systematic errors. Their spectra are similar: in particular both exhibit a strong 557.7 nm [OI] green emission line, whose intensity is used for classification of auroral strength. Unique signature features of each phenomenon are those due to OH band emissions in the case of airglow and $N_2$ (first positive system) emissions in the case of aurora (Chamberlain, 1995). The emission line intensities are usually expressed in Rayleigh (R) units (effectively a measure of directional panchromatic radiance, as per Baker (1974)), with the airglow exhibiting typical 557.7 nm (line-integrated) values of $\sim 0.25$ kR. The International Brightness Coefficient (IBC) is employed to discriminate four aurora classes: IBC1 = 1 kR (brightness of the Milky Way); IBC2 = 10 kR (brightness of thin moonlit cirrus clouds); IBC3 = 100 kR (brightness of moonlit cumulus clouds); and IBC4 = 1000 kR





(provides a total illumination on the ground equivalent to full moonlight) (Chamberlain, 1995). We note that the assessment of the accuracy errors for those classes, may help to infer the effect of moonlight and moonlit clouds too.

Figure 15a shows their emission density spectra (Rayleigh per unit wavelength), converted to 8.2 nm bandwidth of our starphotometers. The airglow data (the black solid curve) represent tropical nighttime observations made by Hanuschik (2003).

These include zodiacal light (sunlight scattered by dust from the solar system ecliptic plane). The actual airglow emissions should accordingly be even weaker. The aurora density spectra (the colored solid curves) are a compilation of observations from Jones and Gattinger (1972), Gattinger and Jones (1974), Jones and Gattinger (1975) and Jones and Gattinger (1976). Their spectra were adjusted to produce three curves, representing the first three IBC levels. Their common continuum (without respect to aurora class) is adjusted to 8 R/Å, the minimum value proposed by Gattinger and Jones (1974).

Figure 15b enables an appreciation of airglow and aurora effects on starphotometer measurements. It shows the ratio of those spectra to the Vega spectrum (artificially attenuated to magnitude $V = 3$, the faint limit of our star dataset). The resulting estimates of optical depth error (equation (30) converted to observational error $x\delta_\tau$ of equation (31)), in the presence of uncorrected emission contributions, correspond to the throughput of the C11. Optical depth errors for the M703 (shown only for the IBC2 case of red dots) are the result of the M703 (FOV-filling IBC2) flux being 2.4 times larger than that of the C11 (i.e.

the ratio of their solid angles, $(57.3/36.9)^2$). We note that, in spite of the fact that the C11 emission spectra are significantly higher in the near IR spectral region, they are, except for the IBC3 case, generally less than 1%. Short term airglow variability induced by air density fluctuations engendered by gravity waves may occur (Nyassor et al., 2018). Figure 15b indicates however that typical airglow conditions have negligible error contribution. Even at twilight, when the Sodium emission lines, at 589.3 nm, can be enhanced by a factor of 5 (i.e. the "Sodium flash" reported by Krassovsky et al. (1962), the potential accuracy error

remains negligible.

On the other hand, the aurora is characterized by a much higher temporal and spatial variability (Dempsey et al., 2005). Beyond that, the aurora shown in Figure 15 is of the green type (i.e. main visible line at 557.7 nm), but one may have other types too, the most common being red, with the main visible line at 630 nm. Therefore, one may get spectral variation too. Such variations may induce significant departures from the nominal emission background spectra of Figure 15a. Considering

the results of Figure 15b the worst estimation of those variations, the optical depth error remains well below 0.01 for the C11 telescope, even when observing a weak $V = 3$ star during an IBC2 aurora (solid red line). An IBC3 Aurora can, given that a (factor of 10) IBC class change is equivalent to a magnitude change of 2.5, be accomodated by employing a sufficiently bright star: the IBC3 representation for a $V = 0.5$ star will decrease to the red (sub 0.01 error) IBC2 curve in Figure 15b. Fortunately, given the current location of Eureka in the auroral oval (Vestine, 1944), IBC3 aurora will only be seen occasionally near the

horizon. Therefore, the accuracy errors of Figure 15b will only appear at airmasses above 5. However, this may change in the next decades, given the recent fast pace migration of the magnetic pole (Witze, 2019; He et al., 2020).

The IBC definition provides a way to also infer $\delta_F/F$ errors associated with the presence of thin moonlit clouds by simply arguing that the red IBC2 curve of Figure 15b also applies to the IBC2 analogy of "thin moonlit cirrus clouds". By definition, the $\delta_F$ spectrum for such a case corresponds to the IBC2 radiance of Figure 15a. The $F$ value for a $V = 0$ star in a thin-

cloud atmosphere can be modelled, in an order of magnitude fashion, by assigning a value of $\tau x = 3$ to equation (21). Using





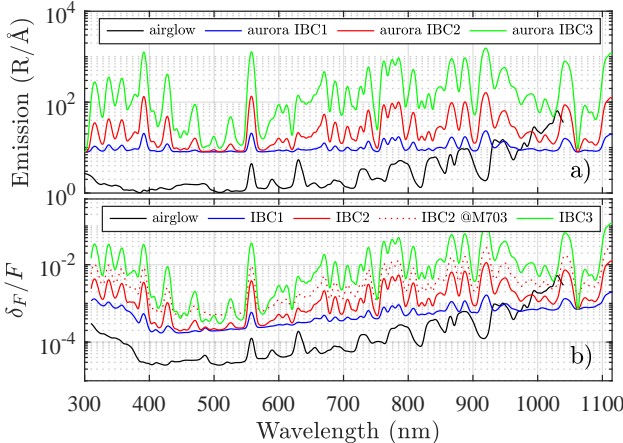

**Figure 15.** Typical emission density spectrum for airglow and aurora (a). Corresponding optical depth error in the presence of uncorrected emission contributions (b): $\delta_\tau = \delta_F/F$ (from equation (30) for $m = 1$), representing the ratio of emission to a Vega spectrum dimmed to $V = 3$ and attenuated in a Rayleigh atmosphere. When observing a $V = 0.5$ star, the corresponding aurora IBC types can be one class brighter to achieve the same optical depth errors. The red-colored dots show comparative $V = 3$ results for the (larger FOV) M703 instrument (see text for details).

this attenuated star signal as a rough model for the IBC2 moonlit clouds analogy, we employ the same equation to show that the $V = 0$, cloud-attenuated star magnitude is equivalent to an unattenuated ($\tau = 0$) $V = 3$ star. In other words, the same $F$ is used to obtain the red $\delta_F/F$ curve of Figure 15b but, with the added rider that the exoatmospheric star was a $V = 0$ star. Accordingly the acceptability of the sub $10^{-2}$ red error curve in Figure 15b applies to the moonlit cloud IBC2 analogy, but for a $V = 0$ star. Actually, given the strong snow albedo in the Arctic, thin cloud brightness may even exceed IBC2 brightness during full moon conditions. Quantitative assessment of optical depth errors related to moonlit and twilight lit sky brightness, especially in cloudy situations, would however require the development of a radiative transfer model informed by starphotometer background measurements. Given the complexity and specificity of such endeavour, this will be addressed in a future study.

The typical polar wintertime night sky background spectrum at Eureka (in terms of catalog-photometric-system magnitude per square arc-second) is shown in the Figure 16a, at two different times: mid-day (magenta curve, local time) and evening (blue). The evening sky is darker and approaches the detection limit of our instrument (as made evident by its noisier profile). This detection limit may be the reason for the difficulty in identifying the aforementioned aurora and airglow lines in the visible range. An omnipresent weak line, unassociated with any major aurora emission lines, is however noticeable around 440 nm (436–445 nm band). Some absorption lines can also be identified: 532 nm, probably due to $O_4$ (Orphal and Chance, 2003) and 663 nm, probably due to $NO_3$ (Orphal et al., 2003). The midnight sky is expected to be even darker (higher visible magnitude). One also notices a brighter infrared spectrum, rather constant throughout the day, confirming the $J$ band measurements of Sivanandam et al. (2012). This may be associated with the airglow $OH$ lines, but a factor $\sim 10$ higher than





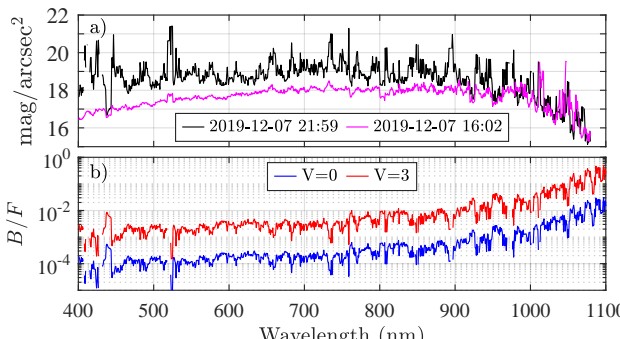

**Figure 16.** a) Night sky background spectrum, measured with the Eureka SPST09/C11 (in the Pulkovo catalogue photometric system), at mid-day (magenta curve) and evening (black curve) during the polar night. b) Ratio of background to star flux, for the evening sky and for two star magnitudes: Vega at $V = 0$ (blue curve) and a dimmed Vega at $V = 3$ (red curve). Times are UTC.

estimated in Figure 15b. The evening sky background with respect to a magnitude $V = 3$ star (simulated by dimming Vega) exceeds the 1% mark beyond 900 nm (red curve in Figure 16b). With respect to a magnitude $V = 0$ star (Vega, blue curve), the evening sky background remains below the 1% mark in the starphotometer spectral range, i.e. < 1050 nm. This indicates that accurate measurements, in the case of weakly radiating ($V > 1$) stars, can only be achieved by applying a reasonably accurate background subtraction for wavelengths larger than 1000 nm.

## 7    Calibration parameter ($C$) accuracy

The accuracy of the calibration parameter $C$ retrieval is dependent on the performance of the calibration procedure and will accordingly be addressed in a separate study. $C$ accounts for the optical and electronic throughput: we asses here the instrument instability or degradation that may alter it.

### 7.1    Misalignment issues

One way to get throughput degradation is by losing flux outside the boundary of the FOV, due to focusing error (blurring), to off-axis star centering errors, or because the FOV is simply too small (design error). The instrument was originally built for the M703 telescope specifications. The smaller FOV of the C11 telescope (almost half of M703's) is at greater risk of focusing errors, particularly in Eureka, where the star spots are larger (Figure 2). An analysis of the impact of design shortcomings on both instruments is an instructive exercise. Figure 17 illustrates the effect of defocusing the optical train within the context of the associated OD errors (case of the C11 telescope) and of star centering errors (cases of both, C11 and M703 telescopes). The fitted curves, which are well modelled by an $a|s|^b$ equation, are only employed to estimate the error variation for low OD (where the density of measurement points is prohibitively small). For the focusing error, the negative and positive $s$ values mean the star spot shift, in steps of the focusing stage (the adjustable unit that controls the focusing of the star photometer, at





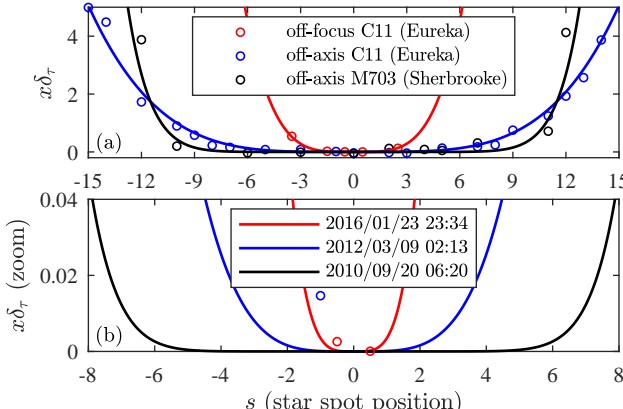

**Figure 17.** Optical depth increase induced by throughput degradation due to misalignment: star focusing error (red for C11) and centering error (blue for C11 and black for M703). For focusing, the positions $s$ represent focusing stage steps (at $\sim 10''$/step increase of the confusion circle); for centering they represent high resolution camera binned pixels, at $3''$/bin for M703 and $2''$/bin for C11. (a) data and $a|s|^b$ fit; (b) the same fit as (a), but zoomed-in to low OD (the measurement date and UTC time is indicated in the legend).

$\sim 1.36$ mm/step along the axis, or equivalently $\sim 10''$/step angular increase of the confusion circle), before and after passing through the on-focus position. For the centering error, they mean the spot shift, in pixels of the high resolution camera, before and after passing through the on-axis position.

Our focusing stage employs a continuously driven motor, subject to electronically controlled steps. Those steps represent approximately the same distance along the optical axis, based on a fixed driving time interval. The best that can be achieved is a half-step focus, for which the flux loss, in the C11 case, is a negligible $\sim 0.02\%$. In the absence of an automatic focusing procedure, the focus has to be checked and adjusted manually whenever there is an important temperature variation. This may happen because of weather changes or as the result of opening the dome (with significant optical impacts up to one hour after the opening). Based on our Arctic experience, the focus must be corrected by one focus step for each $10°C$ change of temperature: if this correction is performed, the flux loss is a negligible $0.35\%$. Any focusing errors larger than that will significantly affect the optical depth estimation (Figure 17b).

Star centering is based on an automatic tracking procedure that ends once a specified centering tolerance $\delta_c$ is satisfied ($\delta_c$ is an input parameter required as part of the starphotometry measuring sequence). Such a tolerance has to be small enough to ensure that, during the subsequent measurement, the star still remains in the accepted centering range, despite any drift due to its natural jitter (spot wandering due to the air turbulence). On the other hand, a faster centering procedure can be achieved using a larger tolerance. There is therefore a trade-off to be made between those two requirements. This is investigated in Appendix C, by taking the constraints posed by the FOV into account. We show, for a perfectly aligned star, that the maximum seeing that the FOV can accommodate is $16.7''$, for our C11 Arctic telescope. This is borderline at $m = 5$ in Figure 2 (long exposure case). Obviously, this somewhat too small FOV is a design shortcoming that can be fixed, for example, with a larger limiting diaphragm.





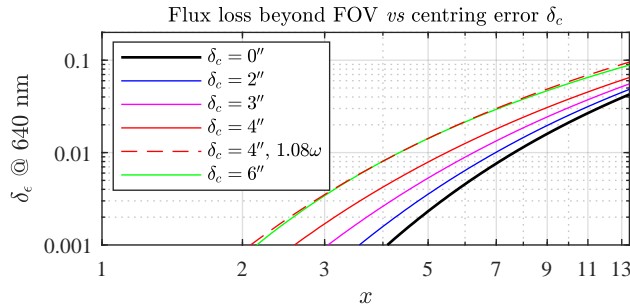

**Figure 18.** Observation error due to throughput degradation in Eureka when using the SPST09/C11 system. This error is the result of a FOV that is too small combined with a centering error (see the main text for details).

In order to asses the accuracy, we employ the calculations of Appendix C to transform the Arctic star spot sizes of Figure 2 into the corresponding observation errors of Figure 18. The black curve represents a systematic throughput degradation due only to the flux loss at the edges of the star spot. Such degradation characterizes the case of a perfectly centered ($\delta_c = 0''$) short exposure star spot ($\omega_s$).

The colored curves account for the attendant error due to different centering tolerance choices ($\delta_c = 2''$, $4''$ & $6''$). We compute them by quasi-quadratically summing (with a 5/3 Kolmogorov turbulence exponent) the natural jitter contribution and the position uncertainty inside the tolerance zone (i.e. $(\sigma_\theta^{5/3} + \delta_c^2)^{1/2}$, with $\sigma_\theta$ from Appendix C). One has to keep in mind, however, that those calculations are based on the blue linear fit of data points used in Figure 2. The possible variations about that line can be estimated inasmuch as the short exposures indicate a standard deviation of 5–10%. This is about the 5% difference between the long and short exposure spot sizes in the Kolmogorov turbulence case, as computed in Appendix C (the approximation $\omega_s \simeq 0.95 \cdot \omega$ following equation (C2)). However, for the purposes of our error modelling, we retained an empirical 8% standard deviation case. The $\omega_s$ values (of a Gaussian distribution in $\omega_s$) may accordingly be greater than $\omega$ values 33% of the time (33% of the Gaussian distribution that extends across the red line of Figure 2 at one standard deviation from its blue-line mean). This $1.08\omega$ case is represented by the dashed red-colored $\delta_c = 4''$ curve of Figure 18. The difference with respect to the plain red curve accounts then for the seeing variation. Since it already exceeds our accuracy limit of 0.01 at $x = 4.4$ (or $m \simeq 5$), it represents the maximum acceptable $\delta_c$ for the constraints of our SPST09/C11 system.

### 7.2 Non-linearity

Non-linearity of detector response to incoming light flux is another source of systematic error. The onset of significant non-linearity conditions occurs at ~8000 cnt/s (i.e. $V = -0.47$ with the C11 telescope, a level normally not reached by any star other than Sirius). If the sky brightness due to atmospheric scattering of sunlight is strong (at dawn or dusk at mid-latitudes, or for longer periods during seasonal shifts of the late and early winter in the Arctic), this limit will be exceeded. The culmination of the non-linearity is that, using our standard 6 s integration, the detector progressively approaches its saturation point at $2^{16}$ counts, or $65535/6 = 10922.5$ cnt/s (i.e. $V = -0.8$ for C11). The consequence, as illustrated in Figure 19, is an apparent de-





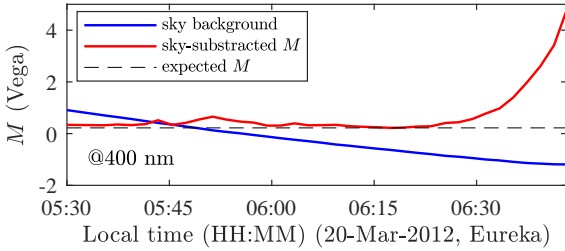

**Figure 19.** Apparent decrease of star brightness (increase of magnitude $M$) as the sky background brings the detector into a non-linear regime. The star brightness has been corrected for sky brightness (the latter has been subtracted out from the former). The separate background measurement (blue line) is not affected by the non-linearity of the detector since $B < 8000$ cnt/s, but the sum $F + B$ used to compute $S$ is, leading to $M = S + C$, equation (13).

crease of star brightness (artificial reduction of the difference between the star and the sky measurements) with a corresponding increase in the computer value of the optical depth. The onset of non-linearity in the case of Vega (whose signal is $\sim 5000$ cnt/s at transit in Eureka) begins at a background ($B$) value of $\sim 3000$ cnt/s (at a total signal of $\sim 8000$ cnt/s as indicated above). One should never employ an instrument such as the C11 to make Sirius ($V = -1.46$) attenuation measurements unless the OD
$> 0.5$. Data whose signal exceeds $8000$ cnt/s should be discarded unless a subtraction process that accounts for the onset of non-linearity is applied.

The sky background is strongly influenced by $O_3$ absorption. This is likely due to the multiple scattering influence of the effective increase in the light path length (from the sub-horizon Sun to the telescope line of sight). This is underscored in Figure 20 where we compare, in a relative fashion, the starphotometer sky background measurements with sky irradiance
(daylight) computations at the bottom of a standard atmosphere for a solar zenith angle of 48.12° "standard indirect solar reference spectrum" (ASTM-G173-03, 2012). The presumed multiple scattering impact of ozone is almost negligible in the latter case, when compared with the starphotometer measurements for the sub-horizon Sun case. One should also note that other absorption bands, like $O_2$ at $\sim 760$ nm or $H_2O$ at $\sim 940$ nm (for example), remain comparable. This means that the non-linearity, as well as saturation, happens first in the blue, leading to a distortion of the retrieved aerosols optical depth spectrum.


### 7.3 Delayed background

Unlike the majority of instrumentally related calibration-degradation influences discussed in this section, the particular problem of delays in background measurements (and the background contamination problem discussed in the next subsection) are of a combined instrumental and observational nature. It concerns bright background conditions, mainly twilight, when the
only feasible observation mode is OSM. If the background subtraction is effected using a background measurement which is delayed in time ($\sim 30$ s) relative to the star measurement (as is the case for our instruments), then $S$ will sustain a systematic error, that becomes progressively worse as the sky brightness increases. Figure 21 illustrates the sky background increase for



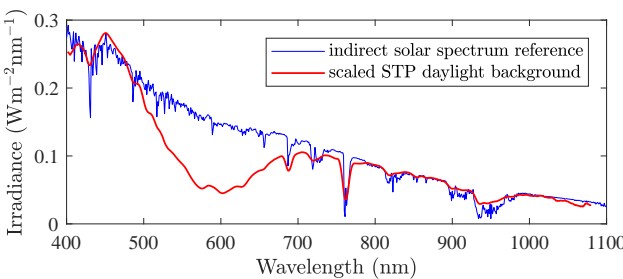

**Figure 20.** Daylight sky background: standard ASTM G173-03 indirect solar reference sky irradiance spectrum (blue); scaled starphotometer background measurements in Eureka (2018-02-18 10:33, local time) to match ASTM infrared level (red).

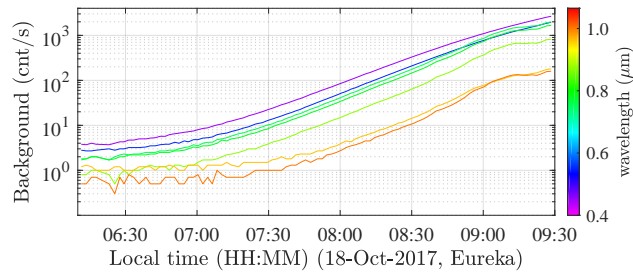

**Figure 21.** Sky brightness increase in the morning, in Eureka, for seven standard channels. The blue part of the spectrum is brightest.

seven standard channels, as acquired at Eureka in support of a morning series of OSM measurements. When those channels (notably the longer wavelength channels of Figure 21) approach the saturation point, near 09:00, the relative rate of increase

$\delta B/B \simeq 0.01$ over the 30 s delay (as computed from the local slope of the curves just before the onset of significant saturation). This leads to an observation error $x\delta_\tau \simeq \delta B/F \simeq 0.01 \cdot B/F$. Accordingly, the sky brightness should never exceed the star's brightness if the OD error is to be less than 0.01. The minimum OD error due to 30 s delay in such anomalously bright (dawn or dusk) conditions is $0.01 \cdot 5000/3000 = 0.017$ for Vega (for other stars is larger, since their $F$ is smaller).

One can nevertheless mitigate this error by extrapolation from outside the saturation regime and correct for it in post-

processing. This procedure is, however, less than ideal inasmuch as the duration spent on a given star measurement is not known precisely due to the unknown duration of the star recentering process between exposures. In any case, one generally expects the residual $\delta_B$ to be 10-20% of the initial. This yields OD errors $< 0.01$ for the entire linear range, even when observing $V = 3$ stars.

### 7.4 Background contamination

Background contamination can also be considered as both, an observation issue and an instrumental issue (i.e. affecting the calibration parameter). This kind of error is a design shortcoming affecting our older instrument versions (SPST05 and 06 which both employ the Losmandy mount). The error has been corrected since it was first noted, but its existence is worth mentioning

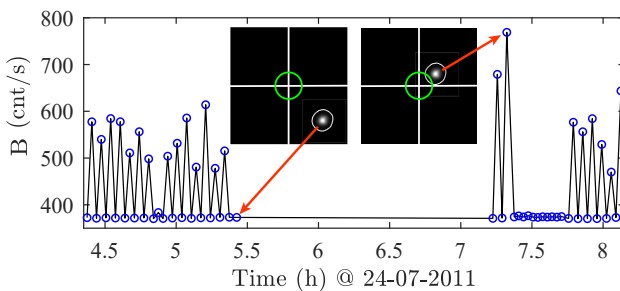

**Figure 22.** Star spot contamination of the background measurements (spikes) observed during the Halifax campaign with SPST05/M703. The spikes occur when background measurements are acquired too close to the star. The 1% star flux level (white circles around the star spots) should never foul the photometer FOV (green circles).

because the source of the problem was not obvious. As indicated above, background subtraction has to be performed subsequent to the star measurement. Based on the Appendix C calculations and on the Eureka star spot sizes shown in Figure 2, the position

of the background measurement should be made at a star separation larger than $35''$ with the C11 telescope and $45''$ with the M703 (i.e. $1.1 \cdot \omega$ at $m = 5$, plus half of the FOV) to make sure the FOV encases less than 1% of the star flux on its border. A separation of $60'' = 1'$ would then be a safe enough margin. A separation of $8'$ was, in actual fact, a feature of the original design (i.e. similar to the FOV of the high resolution camera). However, an oversight in the implementation of that design meant that, for some areas in the sky, the telescope mount fails to achieve the requested move. This can result in erroneous $S$

values induced by the star spot signal contaminating the background measurement. Figure 22 shows one particularly extreme event that occurred during the Halifax campaign (see Section A2 for details of the campaign). Fortunately, we could correct this type of error in post-processing by interpolating between the neighbouring low level, spike-free points on either side of the spikes seen in Figure 22.

### 7.5 Internal temperature variation

The dark current of our detector (S7031-1006 Hamamatsu CCD) varies exponentially with temperature according to the manufacturer's specs. Our instruments incorporate two-stage temperature stabilisation controllers, in order to increase the ambient temperature operation range and accordingly, minimize any temperature sensitive OD retrieval errors. The first stage stabilizes the instrument enclosure to $30 \pm 0.5°C$. The instrument's cold-environment design features include internal heaters to help reach and maintain the temperature set point. It does not however, incorporate coolers to compensate for warmer

temperatures. The influence of warmer temperatures may, as a consequence of the heat generated by the enclosed (quasi-hermetical) electronics, occur when the outside temperature surpasses $0°C$. The only way to cool in such a circumstance is to remove any thermal insulation plates. At higher outside temperatures one simply opens the instrument box for ventilation in open air. The second stage controller is a thermo-electric cooler (TEC), that stabilizes the detector temperature to a standard





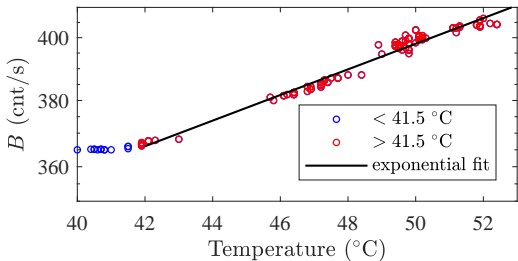

**Figure 23.** Variation of $B$ due to dark current increase with the instrument enclosure temperature above the 41.5°C (1040 nm channel). These measurements were acquired with the SPST05/M703 instrument during the Halifax campaign. The stabilised dark current is 365 cnt/s.

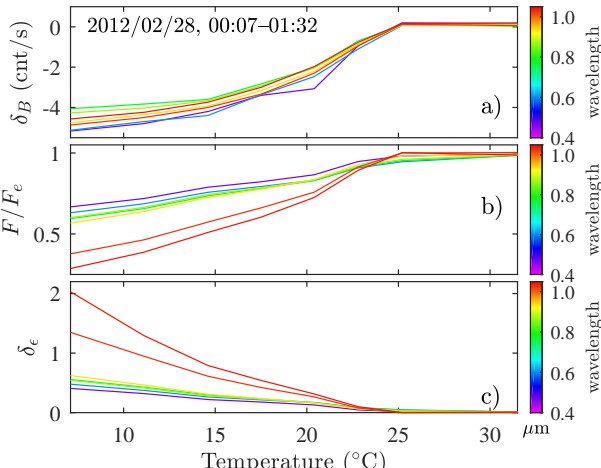

**Figure 24.** $B$ variation ($\delta_B$) due to dark current decrease when the instrument enclosure temperature starts below the control range (a), measured in Eureka with SPST09/C11. Detector sensitivity as a function of temperature (b) and the corresponding observation error $\delta_\epsilon = x\delta_\tau \simeq |F - F_e|/F$ (c). The measurements were acquired at Eureka with the SPST09/C11.

set point of $-10°C$, (adjustable in the $-20°C$ to $-8°C$ range). The TEC can cool down 30° to 45°C below its environment (the instrument enclosure). However, from our measurements, this range is rather found to be 38.5° to 51.5°C.

In warm environments, one can maintain the control up to an enclosure temperature of 41.5°C (Figure 23). Above that, the dark current (the main component of $B$ in dark sky conditions) increases exponentially with the temperature (slightly more pronounced in the near-infrared). In Figure 23 the exponential fit looks linear because of the short vertical range. In cold environments, the instrument enclosure can be subject to temperatures below the lower limit (28.5°C) of its nominal control range. This may happen, for example, during the instrument warm-up phase (Figure 24), or when the outside temperature drops below $-45°C$ and the internal instrument heaters struggle to maintain the $+30°C$ set point. The resulting dark current variation ($\delta_B$) is illustrated in Figure 24a). Because it decreases exponentially with the temperature, its variation is much weaker than that induced by temperatures above the upper limit of the control range. This nonetheless results in significant variation of the





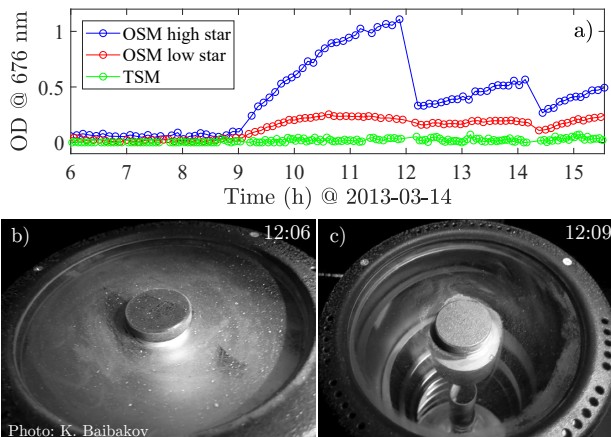

**Figure 25.** Extreme throughput degradation event caused by frost formation on the telescope corrector plate (Barrow campaign). The TSM auto-calibrating mode, effectively compensates, in real time, for any common-mode throughput degradation (attenuation increase) and accordingly remains largely unaffected by the frost (open green circles on (a)). On the other hand, the OSM mode is very sensitive to the apparent attenuation of the high and low stars that constitute the TSM pair (blue and red open circles on (a)). The two rapid decreases in the OSM ODs correspond to cleanings at 12:09 at and 14:20. The two photos, (b) and (c), show the collector plate just prior to and just after the cleaning at 12:09.

detection sensitivity ($F/F_e$), as shown in Figure 24b (where $F_e$ is the star signal once the temperature reaches the nominal control range). This sensitivity ratio is approximately linear with temperature. The much larger slope in the near-infrared channels converts into particularly large observation errors (Figure 24c).

## 7.6 Throughput degradation

Even in the type of clean environment typically found at a mountain top astronomical site, one can notice an optical throughput degradation due to dust deposition on telescope mirrors (Burki et al., 1995). Our starphotometers employ off the shelf (amateur) telescopes, with an optical corrector plate sealing off the optical train and being the main contact surface for any particle deposition. The formation of dew, frost or the deposition of clear-sky snow crystals on that plate represents our greatest source of throughput degradation. Of particular concern is that humidity trapped inside the sealed telescope tube leads to dew or frost formation on the inside of the corrector plate (a degradation which cannot be easily removed by mechanical means). A dramatic event of frost formation, that occurred during the Barrow campaign, (in the absence of a dome or dew cap to protect the telescope), is illustrated in Figure 25. The auto calibrating TSM (c.f. Section 3.4.2) used to derive the green OD points, shows little variation. This indicates that there was likely no aerosol and/or significant cloud OD variation during that period. However, the computed OD associated with the individual high and low stars varies strongly and is, based on our photographic evidence, attributable to frost formation on the plate. One should note that the ramping effects in the OD plot result from progressive frost formation and growth, after two separate damp-cloth cleanings. This operation did not apparently remove all



the frost (the OD values at the beginnings of the two ramps are higher than those acquired prior to the 09:00 time stamp). One should also note that the low star measurements (red data points) are less affected by the frost: this is because the throughput error, as represented by the OD variation from the baseline, is, as per equation (30), divided by a larger airmass.

While at mid-latitudes one usually uses a dew cap to avoid fogging the optics, in the Arctic it cannot be used since it becomes a container for accumulating snow flakes and renders their mechanical removal difficult. One can usually sublimate
the snow and frost from the external side of the corrector plate by closing the dome and increasing the dome temperature by few degrees. However, this doesn't represent the necessary real time solution for preventing throughput degradation. In addition, it doesn't remove any internal telescope frost. Other experiments with limited success were described in Ivănescu et al. (2014). It seemed a rather impossible issue to solve, but a working solution was nevertheless identified, addressing both the frost and incoming crystals: a Kendrick Astro system using a controlled heating band wrapped around the telescope tube. It increases
the temperature of the optics, particularly the corrector plate, by up to 10°C with respect to the environment. One expected this to increase the blurring of the star spots due to micro-turbulence near the telescope, but such effect turned out to be negligible for our instruments.

## 8 Toward 1% accuracy

A relative photometric error of 1% in $\delta_F/F$ represents, in turn, a magnitude error of $\delta_S \simeq 0.01$ and an observational error
of $\delta_\epsilon = x\delta_\tau = 0.01$. We seek to achieve the $\delta_\epsilon < 0.01$ required accuracy goal discussed in the introduction, by mitigating the non-negligible systematic uncertainties identified in this paper.

### 8.1 Optimum channel selection

Some of the largest accuracy errors in starphotometry are, as explained in section 4, due to contamination by stellar and Earth atmosphere (also called telluric) absorption lines, photometer spectral drift, bandwidth mismatch between the instrument and
catalog references, as well as airglow and aurora contamination, when present. These errors can be mitigated through a judicious channel wavelength selection. Avoiding the high frequency spectral influences is also a reason for having narrow (< 10 nm) channels. Since remote sensing photometry is historically based on sunphotometry and much influenced by AERONET standards (the latest being Version 3 (Giles et al., 2019)), the World Meteorological Organization (WMO, 2016) recommendations for photometric-based aerosol observation includes AERONET (central) wavelengths. For consistency, one should
endeavor to select at least a few AERONET bands. Sunphotometry is basically starphotometry based on a spectral class G2 star. Such stars have much weaker hydrogen absorption lines than the typical B and A stars of our catalog. Therefore, our channel selection needs to consider the starphotometry reality, with its specific constraints: mainly to avoid hydrogen ($H$) lines and insuring a star brightness (particularly challenging in near-infrared) much larger than the sky background. Also, selecting more channels than the sunphotometers may help to compensate for typically larger starphotometer observation errors. The
process of selecting more channels in starphotometry is facilitated by the fact that the number of channels employed by our (spectrometer based) starphotometers is not constrained by the time consuming constraints of an AERONET type rotating filter





wheel system. In what follows we attempt to create an OD spectrum with the goal of identifying an optimal starphotometry band set in typical conditions. The method is constrained by an eventual fit to measured OD spectra.

The first step in our band selection process was to identify the spectral intervals free of stellar and aurora/airglow line contamination. To this end, we used the extraterrestrial (HST measured) Vega spectrum (also shown in Section 4), at 8.2 nm bandwidth. Since Vega is a spectral class A0 star, its spectrum, strongly influenced by Balmer and Paschen $H$ lines, is among the most affected by stellar absorption (Silva and Cornell, 1992). Inasmuch as the Vega spectrum can be considered the worst case scenario, the systematic errors due to characteristic stellar absorption bands should be weaker for other stars. In order to obtain only the stellar absorption spectrum, we subtracted the continuum obtained by fitting the magnitude spectrum on

off-lines data points. The result shown in Figure 26a was divided by 1.6, to simulate the airmass of an actual star. An IBC2 aurora OD error spectrum with respect to a $V = 2$ star, together with an airglow 10 times larger than that of Figure 15b, was employed to produce the gold "airglow & aurora" curve. The bottom red bars in Figure 26a delineate the spectral intervals to be avoided, where the total of $H$ lines and aurora contaminants are noticeable ($> 0.007$). For realistic estimates of typical ODs for the most important telluric gaseous absorbers, we used laboratory measured spectra. These included the $O_2$ of Rothman

et al. (2009) adjusted to typical Arctic levels (red curve), the $H_2O$ results of Hill et al. (2013) adjusted to a typical wintertime precipitable water vapour value of 0.8 mm over Eureka (purple curve), and the $O_3$ results of Voigt et al. (2001) adjusted to 250 DU (blue curve). We neglected the $NO_2$ contribution, inasmuch as the measurements of Lindenmaier et al. (2011) identified a maximum $NO_2$ column of $5 \times 10^{15}$ molecules/cm$^2$ = 0.19 DU in Eureka in summer-time, representing $\tau_{NO_2} = 0.003$ at $\lambda = 400$ nm, while in wintertime the models estimate it to a much lower value. The cumulative synthetic absorption spectrum

of these component contributions is shown in dotted green in Figure 26a. We employed the local minima of this curve as band placement indicators for which errors in ascribing values to the ensemble of absorption contributions (which one must inevitably do to extract an aerosol or cloud OD) would be minimized. A set of 20 new channels (solid black vertical lines) was identified as a potential replacement for the old set of 17 channels (dashed grey vertical lines) currently employed in our starphotometers (it also represents approximately 3 times the number of channels employed in the AERONET instruments).

The dotted green curve of Figure 26b shows the same dotted green cumulative spectrum of Figure 26a to which aerosol scattering has been added. The aerosol scattering OD was assumed to vary as per the classical Angstrom expression of $b\lambda^{-a}$, while $b$ was incrementally perturbed until it matched an actual OD Vega spectrum (blue curve) measured at Eureka (a typical value of 1.3 was assumed for $a$). For reference, the same spectrum but without the stellar components is represented in purple. The position of the 20 new channels are duplicated on Figure 26a in order to better appreciate the final total OD context for

those positions.

The selection procedure identifies as many channels as possible, constrained by the avoidance of any absorption line contamination. The ultimate goal is the characterization of the low frequency (slowly varying) aerosol and cloud scattering spectrum. Since there are large spectral intervals where that is not possible (mainly across the $O_3$, $O_2$ and $H_2O$ absorption bands), one also needs to include channels that independently facilitate the extraction of $O_3$ and $H_2O$ column abundances (at least two

channels per band, as they are noisier due to the strong absorption). The newly identified central channel wavelengths, as well as their application and their reason for selection, are summarised in Table 2.




**Table 2.** Specifications for the 20 starphotometry channels chosen according to the absorption feature avoidance process outlined in the text (see the text for details on the reason(s) for selection).

| # | Nominal $\lambda$(nm) | | Application | Reasons for selection |
|---|---|---|---|---|
| 1 | 402 | 401.8 | fine-mode[a] | off $H$ Balmer |
| 2 | 423 | 422.6 | fine-mode | off $H$ Balmer |
| 3 | 446 | 445.9 | fine-mode | $O_3$ base, AERONET |
| 4 | 467 | 466.7 | fine-mode | off $H$ Balmer |
| 5 | 500 | 500.3 | $O_3$[b], fine-mode | WMO & AERONET |
| 6 | 532 | 532.1 | $O_3$, fine-mode | lidar $\lambda$ |
| 7 | 549 | 548.7 | $O_3$ | extra sampling |
| 8 | 595 | 595.3 | $O_3$ | mid twin-peaks |
| 9 | 614 | 614.2 | $O_3$ | extra sampling |
| 10 | 640 | 640.1 | $O_3$ | extra sampling |
| 11 | 675 | 675.2 | $O_3$ | WMO & AERONET |
| 12 | 711 | 711.0 | $O_3$, coarse-mode | extra sampling |
| 13 | 745 | 745.0 | coarse-mode | $O_2$ & $O_3$ baseline |
| 14 | 778 | 778.2 | coarse-mode | WMO $\lambda$ |
| 15 | 845 | 844.8 | coarse-mode | WMO $\lambda$ |
| 16 | 879 | 879.0 | coarse-mode | $H_2O$ base |
| 17 | 936 | 935.7 | $H_2O$ | main peak, off $H$ |
| 18 | 938 | 937.9 | $H_2O$ | mid twin-peaks, off $H$ |
| 19 | 989 | 988.9 | coarse-mode | $H_2O$ base |
| 20 | 1020 | 1020.2 | coarse-mode | AERONET $\lambda$ |

[a] Spectral region that is more sensitive to the characterization of fine-mode (FM) aerosol properties such as FM aerosol OD. The total aerosol OD (FM OD + coarse mode OD) will be sensitive to the presence of FM aerosols.

[b] $O_3$ absorption is sufficiently strong to provide a retrieval of $O_3$ columnar abundance and thus $O_3$ OD from a spectrally dependent matching type of total OD retrieval and accordingly to correct (eliminate) the $O_3$ OD from the total OD for all $O_3$-affected channels.

The justifications for the 20 selected channels (sequentially ordered as per Table 2) are given below:

1. Avoidance or minimization of $H$ contamination. It better constraints the UV/blue trend of the fine-mode aerosol spectrum.

2. Avoidance or minimization of $H$ contamination. This is the optimum $\lambda$ identified in Section 4.

3. Avoidance of an $H$ line, but also the 440 nm emission line identified in section 6.4. This band is near the 440 nm AERONET channel and can be used as the lower bound baseline for isolating the $O_3$ OD band.

4. Avoidance of an $H$ line. Both 3 and 4 channels are moved a bit left with respect to the current (old) channels, to increase their sensitivity to aerosols.

5. WMO recommendation and an AERONET channel.





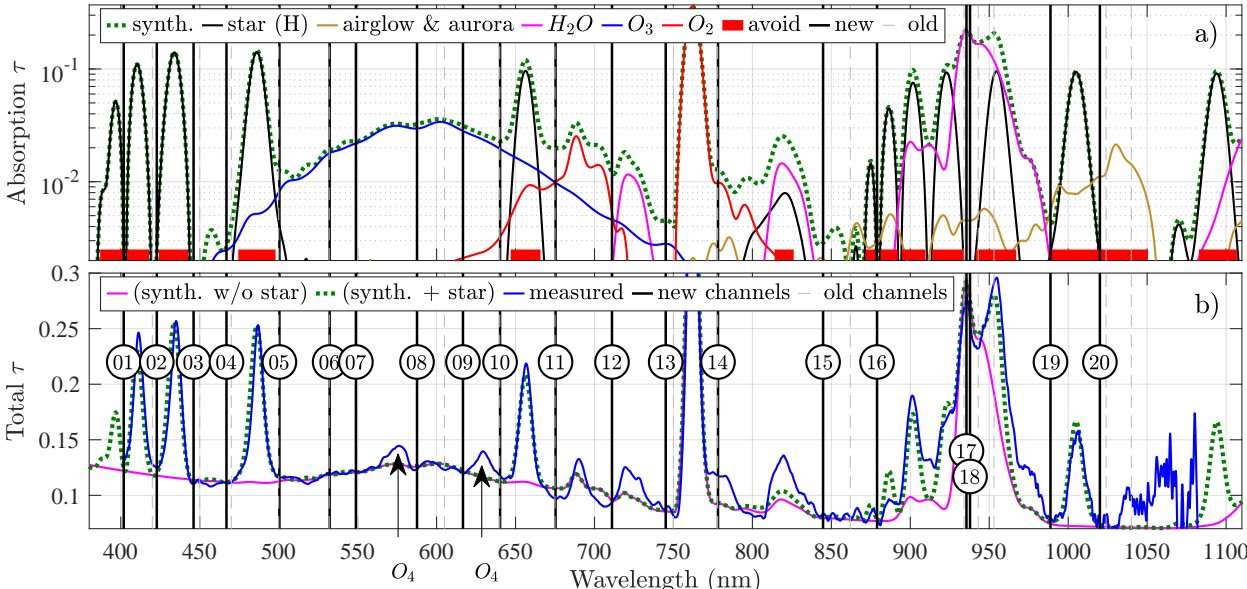

**Figure 26.** OD spectra of constituents that contaminate the retrieval of aerosol and cloud ODs in the visible and near-infrared. a) Starphotometer channel selection (vertical black lines) obtained by avoiding contaminants, such as the stellar (Vega) absorption OD spectrum (black curve), OD errors associated with airglow and IBC2 aurora (gold curve) as well as $O_2$ (at least those parts of its red curve whose $OD \simeq 0.01$). The red bars delimit intervals where those contaminants are non-negligible ($\tau > 0.007$). The channel selection also includes strategically selected regions of $H_2O$ and $O_3$ absorption that allows for the dynamic identification and characterization of their OD and subsequently, their removal from the total OD spectrum. The cumulative contaminant optical depth yields the total synthetic curve (green dotted curve). b) The synthetic curve (with an added aerosol scattering component) versus an OD spectrum (blue curve) retrieved from Vega measurements over Eureka (measured on 2019/11/03 14:01:02). The optimal fit shows generally good agreement except where the contaminant influence is misestimated. This is particularly true for $O_4$ absorption which we realized, a posteriori, should have been included in the ensemble of contaminants. The numbers of the selected channels are superimposed for reference purposes.

6. Lidar standard channel.

7. Good channel for sampling the ozone profile shape while avoiding the 557.7 nm aurora and airglow peak.

8. We note that the difference between the measured and synthetic curves around 590 and 640 nm underscores what appears to be a shortcoming in our synthetic curve: the presence of significant $O_4$ absorption features (see Wagner et al. (2002) for information on $O_4$ absorption). Strong $O_3$ OD channel that lies between double ozone peaks. It's one of 3 bands sensitive to $O_3$ abundance (and thus $O_3$ retrieval). It avoids side-bands of $O_4$ contamination and any possible twilight contamination by 589 nm *Na* flashes (Chamberlain, 1995).

9. Strong $O_3$ OD channel that also avoids an $O_4$ line. Same mandate as band 8 (sensitive to $O_3$ abundance).

10. Avoidance of $O_4$ and $H$ lines. Useful spectral placement for characterizing the $O_3$ profile shape. Requires correction for $O_2$ contamination (note the marginally significant strength of the $O_2$ OD in Figure 26a).





11. WMO recommendation and an AERONET channel. Requires correction for $O_2$ contamination.

12. New channel that fills what, up to this point, has been a large spectral gap in atmospheric photometry. Avoidance of a nearby $H_2O$ line. Requires correction for $O_2$ contamination.

13. Avoidance of the water vapour line at $\sim 840$ nm and the strongest $O_2$ line at $\sim 761$ nm. This channel can be used, respectively, as a lower- and upper-bound baseline for $O_2$ and $O_3$ absorption profiles.

14. WMO recommendation channel. Avoidance of the 840 nm $H_2O$ line and the 761 nm $O_2$ line. Requires correction for $O_2$ contamination.

15. Avoidance of the 840 nm $H_2O$ line. Requires correction for $O_2$ contamination. This channel is near the WMO 862 nm recommendation, but the latter may be affected by aurora OD errors.

16. Avoidance of $H_2O$ and $H$ lines. This channel is meant to serve as a lower-bound baseline for the broad $H_2O$ absorption profile that starts around 890 nm.

17. Maximum $H_2O$ absorption (free from $H$ contamination): used for the retrieval of $H_2O$ abundance. The choice of a maximum also minimises the influence of the line shape variation as a function of water vapor abundance (Volz, 1969).

18. Second $H_2O$ channel to improve measurement precision in low starlight flux conditions: due to strong $H_2O$ absorption and generally low starlight flux in Near-IR.

19. Avoidance of $H_2O$ and $H$ lines. This channel is meant to serve as a lower-bound baseline for the broad $H_2O$ absorption profile that ends around 990 nm. It also avoids the region of the strongest aurora- and airglow-induced OD errors.

20. Channel at the largest near infrared wavelength that still provides accurate measurements, while avoiding $H$ lines. Even if this channel is relatively sensitive to airglow emissions, it can be considered reasonably reliable for $V = 0\text{--}1$ BA class stars, or bright (and colder) F class stars, such as Procyon, whose near-infrared flux is relatively strong.

The major changes and improvements with respect to the original channel set are: a new 402 nm channel to better estimate the UV attenuation due to fine-mode aerosols; 432 nm channel is optimised for minimal contamination; the ozone absorption profile is over sampled to allow better removal in post-processing; the 953 nm $H_2O$ channel was excluded (see it at the right side peak, on top of the $H_2O$ band, Figure 26a), inasmuch as it is likely influenced by a $H$ Paschen line; the $H_2O$ baseline is better estimated with more strategically selected baseline channels (closer to the limits of significant absorption); the original (persistently noisy) 1040 nm channel was excluded (the high frequency variations seen in the retrieved ODs above approximately 1030 nm in Figure 26b is a symptom of the noisy nature of signals in that region of the near infrared). In order to avoid near-IR airglow one needs only acquire data at wavelengths above 1050 nm: this however results in weaker star flux and the above-mentioned weak signal to noise. Finally, we remind that our channel selection process is optimized for the peculiarities of our starphotometer. Signal to noise considerations aside, the spectral bandwidth is one of those peculiarities : different bandwidths may require slightly different channels.



## 8.2 Starphotometry recommendations

We recommend the general usage of the 20 starphotometry channels defined in the previous section. Those channels are dedicated to the extraction of aerosol and/or cloud ODs as well as the strong molecular absorbers of $O_3$ and $H_2O$ (either as

corrections to achieve estimates of aerosol and cloud ODs or, as remote sensing targets on their own merit).

An important source of OD error is related to the accuracy of the spectrophotometric catalog. In the case of the Pulkovo catalog, we identified a particularly large bias in the UV and 900–1000 nm regions (c.f. the text associated with Figure 4), that could distort the retrieved aerosol spectrum. That bias aside, the errors in the individual star spectra are also prohibitive in terms of achieving the required accuracy. It is strongly recommended that a new and improved bright star catalog should

be made, preferably with magnitude measurements acquired by a space-based instrument, to avoid the incertitude related to telluric absorption contributions. As discussed in Section 4, the requirements for such a catalog are 1 nm bandwidth and $< 1$ nm (preferably 1 Å) spectral resolution, with less than 0.01 differential magnitude variation across the measured spectrum. In the mean time, we continue to use the Pulkovo catalog, but with its 8.2 nm bandwidth version that improves the bandwidth match with our instruments, and offers a wider bright star diversity than what is currently provided by HST. Alternatively, if the

starphotometer is spectrometer-based, as is our starphotometer, then one can generate such a catalog from direct high resolution observations (all spectrometer channels) at a high altitude site. Such a catalog would perfectly match the instrument bandwidth. We recommend, that future starphotometer bandwidths be held to less than 10 nm : this is an easily attainable standard that ensures negligible heterochromatic errors ($\delta_\tau < 0.001$). The employment of all the spectrometer channels ensures that any high resolution stellar features can be properly accounted for in future corrections of any spectral drift (before extracting a drift-

modified set of optimized operational set of starphotometer channels). Observations and calibration should be preferentially performed with a B0–B3 (early-type B) star, or A7–A9 (late-type A) and F stars, to avoid the incertitudes related to the strong stellar absorption lines. Also, annual spectral calibration is advisable in the face of the drift results of Figure 8. Alternatively, measurements of a high resolution spectrum of a particularly bright star of near-A0 type (notably Vega) could be carried out every few months. Its deep $H_{\alpha,\beta,\gamma}$ Balmer lines will serve, together with the Earth's $O_2$ and $H_2O$ lines, as reference for

spectral calibration in post-processing. One particular concern at mid-latitudes locations, is that $NO_2$ may be several times larger (Cede et al., 2006) than Eureka (i.e. up to 0.03 OD at 400 nm) and its absorption will be no longer negligible. Since $NO_2$ absorption is impossible to discriminate from aerosol spectrum, it has to be assessed from independent sources.

Retrievals in the presence of rapid temporal variations of sky brightness (a measurement which must accompany every star measurement) must be corrected by interpolating from pre- an post-contaminated sky brightness measurements to the

time of the star measurement. Signals greater than the threshold for the onset of non-linearity (8000 cnt/s in the case of our starphotometer) should be discarded (Section 7.2). In such bright sky conditions, one may expect OD errors $> 0.017$, unless interpolating the sky brightness to the time-stamps of star measurements. One should be aware that, at sky brightness increase, the blue part of the spectrum saturates first, leading to a distorted aerosol spectrum retrieval.

Airmass accuracy should be ensured by the use of a GPS time server. OD errors associated with airmass uncertainties can

also be reduced at a high altitude site, while they remain sensitive to time errors on low stars, i.e. at large $x$ (c.f. Figure 12). The





internal instrument temperature should be monitored, inasmuch as the temperature controller may eventually fail (for example, at the very low environmental temperatures found in the Arctic). One particularly needs to wait for the system to warm up to its stabilised range, as the low temperatures have a larger error impact (c.f. Section 7.5).

The stability of the throughput has fundamental impact on the calibration process. Due to the excessively small FOV of the SPST09/C11 configuration ($36.9''$), the optical alignment proved to be critical to ensuring stable throughput. As demonstrated in the discussion surrounding Figure 18, the centering tolerance error should not exceed $4''$ for this instrument in Eureka (2 CCD bins). The focusing error of SPST09/C11 should always be within one step adjustment step (confusion circle variation of $\sim 10''$/step, as per the legend of Figure 17). This means, for example, that the focus must be adjusted by one step for each 10°C change in outside temperature.

Turbulence analysis using star spot imaging revealed another large source of throughput degradation, which is acerbated at Arctic sea-level sites: possible vignetting of star spots at large airmasses. This problem was ascribed to the small SPST09/C11 FOV in the context of the excessively larger seeing of Eureka. The worst case ($m = 10$) scenario of the red curve in Figure 2 (for which the star spot is $\omega \simeq 19''$) can be accomodated by a FOV of $2.3\omega \simeq 45''$ (see Appendix C for details). In the light of the forward scattering error analysis, one should not increase it beyond $\sim 47''$ (roughly where the mean of the most demanding

cases of Figure 14, the "120 $\mu$m (ASC) ice cloud" case, crosses the 0.01 value of the $\delta_\tau/\tau$ axis). This FOV limitation also ensures accurate measurements (sub 1% errors associated with the brightness contamination cases of Figures 15 and 16) during faint aurora (IBC2) events (or their illumination-equivalence of thin moonlit cirrus clouds) for even weak ($V = 3$) stars (with the near infrared exception of Figure 16 where a bright, $V = 0$ star such as Vega, is needed to achieve the 1% threshold). A $45''$ FOV, which would be obtained with a 0.61 mm diaphragm in the C11 case, therefore appears to be good compromise

between the conflicting requirements of maximizing the FOV to accommodate all star spot sizes (red curve of Figure 2) and limiting the FOV to minimize the largest forward scattering errors (blue curves of Figure 14). The small FOVs employed in starphotometry ensure that this technique is significantly less dependent on the intrinsic and artificial OD reduction induced by scattering into the FOV by optically thin clouds. This singular capability of starphotometry renders it rather unique in extinction-based photometry inasmuch as sun- and moon-based techniques require (or at least traditionally use) much larger

FOVs and accordingly suffer from much larger FOV scattering contamination.

We demonstrated (Section 7.1) that observations at airmasses higher than $\sim 5$ should not be made with the C11 because of the influence of vignetting. Calibration may nevertheless be performed beyond this airmass limit, as long the $S$ values still show a linear dependence on $x$. This may happen in weaker air turbulence conditions than those of Figure 2. Throughput degradation due to frost/dew or ice crystals deposition on the telescope was a longstanding problem of our Eureka starphotometer (with

critical accuracy implications). The use of the Kendrick system (or similar heating bands), together with a small wind shield, proved to be a reliable solution which would be appropriate for most of Arctic observing sites. If all these recommendations are followed, one may aspire to achieve a reduction of each zenith OD error component to well bellow 0.01 and the total zenith OD error to $\lesssim 0.01$ (i.e. the stated 1% photometric accuracy). Even if these goals are, in certain cases still under development, any progress that substantially approaches the goal of 0.01 total zenith OD error would represent a significant advance in

starphotometry reliability.





## 9   Conclusions

With the ultimate goal of improving starphotometry accuracy, we analysed a large variety of sources leading to systematic (absolute) errors and classified them by their impact on each parameter involved in the optical depth retrieval. The contamination from stellar and telluric gas absorption lines may potentially induce large OD errors. One of the newly identified contaminants are $O_4$ absorption lines, that affect $O_3$ estimation and removal, leading to distorted aerosol OD spectrum. Such errors are nevertheless mitigated with proper channel allocation: this we demonstrated using synthetic and measured OD spectra to extract a set of 20 optimal channels. In order to minimize further the absorption lines induced OD errors (stronger hydrogen lines tend to spill over into different bands), one may favour the starphotometry observations using early-type B, late-type A and F spectral class stars, that have weaker hydrogen absorption lines. Therefore, we may particularly prefer them for calibration purposes.

Inaccuracies in the current exoatmospheric photometric catalog can be partly addressed in the TSM observation mode (where the catalog bias is cancelled out) or by circumventing the catalog with lengthy calibrations involving each star that one wishes to employ as an extinction target (calibrations using Langley calibrations at a high altitude site, for example). Given such restrictive options, the community is strongly encouraged to prioritize the development of a new spectrophotometric catalog with improved accuracy, supported by magnitude variability characterization. This will increase confidence in the accuracy of a star independent calibration, and render that approach more operational and reliable.

Problems related to the instrument instability (including spectral drift and star spot vignetting) were identified and appropriate observation strategies and design improvements were proposed. Beyond the current accuracy assessment study, we will pursue starphotometry reliability improvement by also characterising the non-systematic, random errors, as well as those related to $C$ values retrievals through Langley plot calibration. A sky brightness model to estimate the background of moonlit and twilight lit clouds is in development. A new exoatmospheric photometric catalog based on GOMOS satellite photometry is also envisioned. In order to validate the proposed improvements, one should participate in observing campaigns and compare the observations with other collocated instruments. The CIMEL moonphotometer and the profiling backscatter lidar at our Eureka site are collocated instruments that already provide support of this nature.

As an original by-product of this study, we developed a semi-empirical expression for estimating the seeing (star-spot blurring) profile from radiosondes measurements.

*Code and data availability.*   The Matlab code and data employed in the generation of the figures are freely available (Ivănescu, 2021).

## Appendix A:  The Canadian starphotometry program

Our group at the Université de Sherbrooke has been performing starphotometry observations of aerosols and optically thin clouds in Canada and elsewhere since 2007. There have been a total of three Canadian sites in our small starphotometer network (the high latitude site at Eureka, Nunavut, and mid-latitude sites at Sherbrooke, Québec and Egbert, Ontario). Currently the





network has been reduced to the Eureka and Sherbrooke sites. Additionally, campaign-based observations took place in Halifax (NS), Barrow (Alaska, USA) and Izaña (Canary Islands, Spain).

## A1  The instruments

Our starphotometers were built by Dr. Schulz & Partner GmbH, a German company that has now ceased operations. A total
of 9 instruments, with serial numbers SPST01 to SPST09, were produced. The first three were initial-development versions (now decommissioned) and the remaining six are still in operation. Three of these are German owned: SPST04 is still at the manufacturer, SPST07 and SPST08 are being operated, respectively, at the Lindenberg observatory in Germany and by the Alfred-Wegener Institut (AWI) at Ny-Ålesund (Svalbard). The other three are Canadian owned: SPST05 is at the Université de Sherbrooke while SPST06, formerly at the Egbert Environment and Climate Change Canada (ECCC) site, has been decom-
missioned. SPST09 is at the Eureka OPAL site (Ivănescu et al., 2014). The SPST04 to SPST07 instruments are all of the same version, while the two most recent versions, SPST08 and SPST09, are upgrades. The common detection device employed for all those versions is the QE65000 scientific-grade spectrometer from Ocean Optics. The QE65000 is based on a Hamamatsu S7031-1006 CCD sensor ($1044 \times 64$ pixels). We use two different telescopes, both having an $f_\# = f/D = 10$ focal ratio, where $f$ is the focal length and $D$ the diameter.
The telescope "plate scale" $P_s$ on the focal plane, can be computed (Carroll and Ostlie, 2007) with

$$P_s = k_c/f = k_c/(D \cdot f_\#)$$

where $P_s$, having units of $['' /\mathrm{mm}]$ with $k_c = 3600 \cdot 360/(2\pi) = 206264.8$ $''/\mathrm{rad}$, is a radian to $['']$ conversion factor, and $f$ and $D$ are in $[\mathrm{mm}]$. The version to version improvements concern mainly the robustness of the instrument. However, the throughput of the SPST09 instrument is a factor of 3.2 better than the previous version due to the use of an 11-inch-diameter
(279.4 mm) Celestron C11 Schmidt-Cassegrain telescope with Starbright XLT coating and a 98 mm diameter (secondary mirror) central obstruction (11.5% of the primary mirror surface). The previous models used a 7 inch (177.8 mm) Alter M703 Maksutov-Cassegrain telescope with a 32% (secondary mirror) central obstruction (100 mm diameter). The internal optics of the SPST09 are currently coated with (Melles Griot) Extended HEBBAR$^{\mathrm{TM}}$ coating. By comparison, all other starphotometer versions have custom coatings with about 3 mag. throughput loss around 500 nm, but with about one magnitude gain in the
infrared. All versions perform measurements simultaneously across 1000 channels along the 1044 pixels of the CCD: only 17 (multi-pixel) bands were selected by the manufacturer as a standard for regular operation (see Table 1). Near-star, night sky radiance for background subtraction from the stellar signal is measured by pointing the photometer about $8'$ (arc-minutes) off-target. The star-acquisition procedure is based on star centering by two auxiliary SBIG ST-402ME-C2 CCD cameras. A square $504 \times 504$ pixels (px) sub-frame of the available $510 \times 765$ px CCD frame is employed. For speed and sensitivity, the
acquisition mode uses $3 \times 3$ bins of 9 $\mu$m square pixels (i.e. $27 \times 27$ $\mu$m bins). The initial wide-field centering uses a 67 mm diameter refractive auxiliary telescope with a fast $f_\# = 4$ focal ratio. Its $P_s = 12.65'/\mathrm{mm}$ (arc-minutes per mm) plate scale provides a $57.4'$ field of view (FOV) on its camera, with $20.5''/\mathrm{bin}$ (arc-seconds per 3 pixel bin). The subsequent centering is done at high angular resolution, using the main telescope. The $P_s = 73.7''/\mathrm{mm}$ plate scale of the C11 telescope provides





a 5.6′ FOV, at 2″/bin. The $P_s = 114.6″$/mm plate scale of the M703 telescope provides a 8.3′ FOV, at 3″/bin. Based on the Nyquist-Shannon sampling theorem (Shannon, 1948), one can track star spots at the maximum precision of 1 bin if one has at least one bin per standard deviation of the star spot (Robertson, 2017), or 2.355 bins per Full Width at Half Maximum (FWHM). This would be the case for star-spot FWHMs larger than 4.7″ for the C11 and 7.1″ for the M703. This condition is easily satisfied for the C11, but only satisfied for $m \gtrsim 2.5$ in the case of the M703 (c.f. Figure 2) . To avoid contamination from off-target objects, one limits the measured FOV with a 0.5 mm diameter diaphragm, at the telescope focus. This means, based on the corresponding plate scale, that the spectrometer (i.e. the actual detector) FOV is 36.9″ for C11 and 57.3″ for M703. The star light is then refocused on a 400 $\mu$m diameter optical fiber, which feeds the QE65000 grating spectrometer through a 200 $\mu$m wide slit. The diffraction profile, on the spectrometer's 1044 pixel-long CCD, covers several (24.6 $\mu$m square) pixels, at 0.7 nm/px. In order to improve the measurement accuracy, one averages 5 pixels ($\pm 2$ around the central pixel). The convolution of the slit function with the averaged pixels leads to a profile having FWHM$\simeq 8.2$ nm, or 12 pixels (the bandwidth reported in section 4). Assuming Gaussian shaped bands, each channel suffers $> 1\%$ contamination from blur within 10 nm from its center. The typical starphotometer measurement implies simultaneously averaging several (usually 3 or 5) 6 second exposures, in all channels. Other technical parameters are listed in Table 1.

All instruments are protected by astronomical domes. There are 12 and 7 ft diameter Astrohaven domes in Sherbrooke and Egbert, while Eureka boasts a 10 m diameter dome built especially for Arctic conditions by the Baader Planetarium in Germany (Figure 1). The tracking system (the telescope mount) at Sherbrooke and Egbert is the Losmandy G-11 German equatorial mount. A AZA-2000 Dobsonian alt-azimuth mount, especially built for the Arctic by the Italian company 10Micron, is employed at Eureka.

## A2 Observing sites

The Sherbrooke, Quebec, site is located within the Université de Sherbrooke campus, on the roof of the SIRENE ("Site Interdisciplinaire de REcherche en ENvironnement Extérieur") measurement station (coordinates 45.374°N, 71.923°W, and ground elevation + instrument height of 308 + 6 m ASL. The Egbert, Ontario, site is at the ECCC "Centre for Atmospheric Research Experiments" (44.232°N, 79.781°W, 251+6 m ASL), located 65 km North of Toronto, Ontario.

The Eureka, Nunavut site (79.991°N, 85.939°W, 10+2 m ASL) which is part of the Zero Altitude PEARL Auxiliary Laboratory (0PAL) site near the ECCC Eureka Weather Station is our most prolific data provider. Polar nightime data, from roughly late September to late March, was acquired from 2008 to 2010 using the SPST05/M703 instrument. The upgraded SPST09/C11 collected data for about two months during each observing seasons until 2014 with a gap in 2012-2013 (Ivănescu et al., 2014). After overcoming several technical difficulties, the acquisition period was extended to 3-4 months from 2015 onwards.

The Halifax site was on the roof of the Sir James Dunn Building (44.638°N, 63.593°W, 45+6 m ASL), at Dalhousie University. Two weeks of data were acquired with SPST05/M703 during the July 2011 BORTAS campaign ("BOReal forest fires on Tropospheric oxidants over the Atlantic using Aircraft and Satellites"). Outside of Canada, we performed SPST06/M703 observations, for about a week, in October 2008, at the "Izaña Atmospheric Research Center" in Tenerife, Canary Islands,



Spain (28.309°N, 16.499°W, 2390+1 m ASL). In March 2013, we carried out a SPST05/M703 field campaign at the Barrow, Alaska Observatory (71.323°N, 156.611°W, 11+2 m ASL).

## Appendix B: Star dataset

Our 20-star selection from the dataset of Northern Hemisphere bright stars is presented in Table B1. These stars were selected for their stability (with the requirement that DE $\gtrsim$ -23.5° to account for Earth axis inclination): the 13 positive DE stars are usually present in the Arctic sky. One can always form a ("High","Low") pair from those 13 stars and thus have recourse to the TSM mode.

The GCVS catalog (the source of the $\Delta$V parameter) is built on old observations dating as far back as 1949, while the Hip-
parcos catalog (the source of the $\Delta$Hp parameter) has a photometric resolution magnitude limit of 0.01 but only intermittently monitors a given star. While $\Delta$Hp $\leq 0.01$ for only a few stars in Table B1, there are 24 such $V < 3$ stars in the Pulkovo catalog. Given the uncertainty in star variability, as evidenced by discrepancies between the $\Delta$V and $\Delta$Hp columns of Table B1, a proposal for a new Table B1 dataset should wait for a more reliable future photometric catalog.

The similar spectral class constraint on pairs of TSM stars (section 4) indicates that the Table B1 pairings should be the (HR
7001, HR 7557) of A class, as well as the (HR 1791, HR 1790) and (HR 5191, HR 3982) of B class. These pairs have similar RA values (meaning that they are fairly close in azimuth) and, together, cover the entire 24 h period (while ensuring airmasses < 6 for low stars). We note that the spectral subclasses differ substantially for all three pairs (i.e. the 0–9 class suffix): however if we loosen the RA criterion then the alternate (HR 5191, HR 1790) pair may be of sufficiently similar subclass.

## Appendix C: FOV constraints

The long exposure PSF is characterized by a FWHM$\simeq \omega$ and a standard deviation $\sigma = \omega/2.355$. A large part of the PSF is due to random star-spot movements, called jitter ($\theta$). The fact that the short-exposure movement is tracked dynamically by the starphotometer means the low frequency jitter ($\theta_L$) is largely reduced and thus will not contribute to the star spot fed into the photometer. We estimate $\sigma_{\theta_L}^2$ by integrating the jitter power spectrum, from 0 up to the tracking bandwidth (i.e. half of the low-frequency value given in equation (15) of Glindemann (1997)). When the tracking bandwidth tends to the sampling
frequency ($1/t$), the missing jitter contribution to $\omega$ is (ibid)

$$\omega_{\theta_L} = 2.355 \cdot \sigma_{\theta_L} = 0.917 \left(\frac{r_0}{vt}\right)^{1/6} \omega \tag{C1}$$

where this equation and all equations in this appendix are homogeneous as a function of angle (i.e. the use of a nonstandard angular argument such as arc-seconds scales coherently on both sides of homogeneous equations). The turbulence length parameter was found to be $r_0 \simeq 0.01$ m for Eureka (section 2) and $v \simeq 10$ m/s is the typical effective wind speed. The operational
starphotometer exposure value of $t = 6$ s yields,

$$\omega_{\theta_L} = 0.215 \cdot \omega \tag{C2}$$





**Table B1.** Star selection from the Northern Hemisphere bright stars. These stars are usually referred to by their Bright Star Harvard Revised (HR) Photometry catalog (Pickering, 1908). Their HD (Henry Draper catalog; Cannon and Pickering (1918)), and HIP (Hipparcos catalog; van Leeuwen et al. (1997)) codes are also listed in order to facilitate their identification. The subsequent columns show their affiliated rank (Greek letter) and constellation, their common name, Right Ascension (RA) and Declination (DE) coordinates at epoch 2000, visual magnitude (V). GCVS and Hipparcos peak-to-peak magnitude variations ($\Delta V$ and $\Delta Hp$, respectively) are indicators of star stability. The next column shows the spectral class (Sp) of the star (including its 0–9 numerical subclass) and its luminosity class (Lum). The last column is specific to the Arctic; it indicates the TSM role of each Arctic star ("High" or "Low"), as described in section 3.4.2).

| HR | HD | HIP | Rank Constellation | Name | RA(2000) | DE(2000) | V | $\Delta V^a$ | $\Delta Hp^b$ | Sp/Lum | TSM |
|---|---|---|---|---|---|---|---|---|---|---|---|
| 15 | 358 | 677 | Alpha Andromeda | Alpheratz | 00:08:23 | 29°05:26 | 2.06 | 0.04 | 0.02 | B8I/Vp | High |
| 1790 | 35468 | 25336 | Gamma Orion | Bellatrix | 05:25:08 | 06°20:59 | 1.64 | 0.05 | 0.03 | B2/III | Low |
| 1791 | 35497 | 25428 | Beta Taurus | Elnath | 05:26:18 | 28°36:27 | 1.65 | - | 0.01 | B7/III | High |
| 2004 | 38771 | 27366 | Kappa Orion | Saiph | 05:47:45 | –09°40:11 | 2.06 | 0.08 | 0.03 | B0.5/Ia | - |
| 2421 | 47105 | 31681 | Gamma Gemini | Alhena | 06:37:43 | 16°23:57 | 1.93 | - | 0.02 | A0/IV | Low |
| 2491 | 48915 | 32349 | Alpha Canis Major | Sirius | 06:45:09 | –16°42:58 | –1.46 | 0.05 | 0.19 | A1/Vm | - |
| 2618 | 52089 | 33579 | Epsilon Canis Major | Adharaz | 06:58:37 | –28°58:20 | 1.50 | - | 0.01 | B2/II | - |
| 2943 | 61421 | 37279 | Alpha Canis Minor | Procyon | 07:39:18 | 05°13:30 | 0.38 | 0.07 | 0.07 | F5/IV-V | Low |
| 3982 | 87901 | 49669 | Alpha Leo | Regulus | 10:08:22 | 11°58:02 | 1.35 | 0.07 | 0.03 | B7/V | Low |
| 4295 | 95418 | 53910 | Beta Ursa Major | Merak | 11:01:50 | 56°22:57 | 2.37 | 0.05 | 0.02 | A1/V | High |
| 4534 | 102647 | 57632 | Beta Leo | Denebola | 11:49:04 | 14°34:19 | 2.14 | 0.025 | 0.02 | A3/V | Low |
| 4662 | 106625 | 59803 | Gamma Corvus | Gienah | 12:15:48 | –17°32:31 | 2.59 | 0.04 | 0.02 | B8/IIIp | - |
| 5191 | 120315 | 67301 | Eta Ursa Major | Alkaid | 13:47:32 | 49°18:48 | 1.86 | 0.06 | 0.02 | B3/V | High |
| 6378 | 155125 | 84012 | Eta Ophiuchus | Sabik | 17:10:23 | –15°43:29 | 2.43 | - | 0.02 | A2/V | - |
| 6556 | 159561 | 86032 | Alpha Ophiuchus | Rasalhague | 17:34:56 | 12°33:36 | 2.08 | 0.11 | 0.02 | A5/III | Low |
| 7001 | 172167 | 91262 | Alpha Lyra | Vega | 18:36:56 | 38°47:01 | 0.03 | 0.09 | 0.06 | A0/Va | High |
| 7121 | 175191 | 92855 | Sigma Sagittarius | Nunki | 18:55:20 | –26°17:43 | 2.02 | - | 0.03 | B2.5/V | - |
| 7557 | 187642 | 97649 | Alpha Aquila | Altair | 19:50:47 | 08°52:06 | 0.77 | 0.004 | 0.05 | A7/V | Low |
| 8728 | 216956 | 113368 | Alpha Pisces Australids | Formalhaut | 22:57:42 | –29°37:01 | 1.16 | 0.01 | 0.01 | A3/V | - |
| 8781 | 218045 | 113963 | Alpha Pegasus | Markab | 23:04:49 | 15°12:38 | 2.49 | 0.05 | 0.01 | B9/V | Low |

$^a$ From General Catalog of Variable Stars: version GCVS 5.1 (Samus et al., 2017). $^b$ From Hipparcos Main Catalog (van Leeuwen et al., 1997).

The FWHM of the $t = 6$ (starphotometer) short exposure spot in a Kolmogorov turbulence is $\omega_s = (\omega^{5/3} - \omega_{\theta_L}^{5/3})^{3/5} \simeq 0.95 \cdot \omega$ or more, depending on the performance of the tracking system. This means that the tracking basically applies a negligible correction to $\omega$. Averages of the Figure 2 ($\omega_s$) points for a given value of $m$ indicate a ratio relative to $\omega$ that is somewhat smaller than the 0.95 implied above. In effect, the preparation of Figure 2 necessitated short-exposure time reductions from the 6 s operational standard in order to circumvent problems such as signal saturation: that figure is more realistic in terms of providing a cross section of short-exposure times that might be used by starphotometers in general.



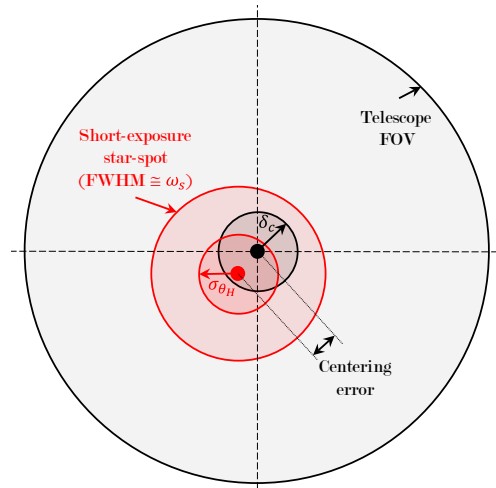

**Figure C1.** Schematic of one possible position and size of the short exposure star spot (bright red) relative to the SBIG-camera CCD-grid (dashed lines), for the case $\delta_c = \sigma_{\theta_H}$. The telescope FOV is shown in black: its center (solid black circle) nominally defines the origin of the SBIG camera grid. We define the "centering error" as the distance between the center of the (black) telescope FOV and the center of the (red) short exposure star-spot.

Based on equations (77) from Tyler (1994) and (5) from Racine (1996), the standard deviation of the total jitter $\sigma_\theta$ is

$$\sigma_\theta = 0.42 \cdot (\lambda/D)^{1/6} \cdot \omega^{5/6} \tag{C3}$$

For our instruments and telescopes $(\lambda/D)^{1/6} \simeq 1''$. One can show that, for $5'' < \omega < 15''$ (i.e. the $\omega$ range at $m < 5$ in Figure 2), one can approximate $\omega^{5/6} \simeq 0.7 \cdot \omega$ and recast equation (C3) as

$$\sigma_\theta \simeq 0.4 \cdot \omega^{5/6} \simeq 0.3 \cdot \omega \tag{C4}$$

Using equation (C2), one retrieves the high-frequency component of the jitter as $\sigma_{\theta_H} = (\sigma_\theta^2 - \sigma_{\theta_L}^2)^{1/2} \simeq 0.21 \cdot \omega$. This represents an $\omega$ dependent estimation of the star spot displacement between the starphotometer measurements. A centering tolerance of $\delta_c = \sigma_{\theta_H}$, specified in the star-centering process (for an assumed Gaussian probability distribution of the random jitter), ensures that about 2/3 of the subsequent, short exposure measurements, will still be centered (see Figure C1 for a schematic of star spot positions and their defining parameters). The ($m = 5$) long exposure $\omega$ values at Eureka and Sherbrooke of $14.7''$ and $8.9''$, respectively (Figure 2), imply a ($\delta_c \simeq 0.21 \cdot \omega = 0.2 \cdot \omega_s$) centering tolerance of $3.1''$ and $1.9''$, or roughly 2 and 1 pixels, respectively. This is consistent with Baudat (2017) $\omega/4$ rule of thumb suggestion of acceptable tracking error.

Figure C2 shows a snapshot of the C11 short exposure tracking process for a high and low star ($4'' = 2$ pixels centering error and 3 choices of centering tolerance). The high and low stars illustrate, notably in the latter ($m = 4.9$) case, the flux loss beyond the FOV boundaries for even short exposure star spots. Using Gaussian distribution calculations and the $\omega_s \simeq 0.95 \cdot \omega$ relationship, one can show that the flux loss will be <1% if $\omega <$ FOV/2.3. This translates to a maximum seeing ($\omega$ value) of



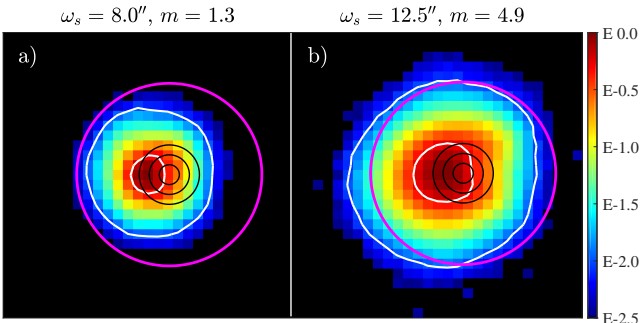

**Figure C2.** SPST09/C11 tracking of a 6 s short-exposure, high-star spot (a) and a low-star spot (b). These two illustrations represent high-resolution SBIG camera data that correspond to two Eureka points on Figure 2, but whose airmasses of 1.3 and 4.9 were obtained from the intersection of their $\omega_s$ values with the blue regression line. The a) and b) fluxes are normalized to their maximum flux (their log-scale colour legend is shown to the right). The FOV, the 1% and 50% flux levels and the centering tolerances are represented respectively by a magenta circle, the 2 white contours and the 3 black concentric circles (radius of 2, 4 and $6''$). The spots are horizontally shifted by a $4''$ centering error w.r.t. the FOV.

$25''$ for the M703 telescope. However, the same calculation gives $16''$ as the maximum seeing that one can accommodate for a perfectly centred star, using the Arctic C11 telescope. This value is problematic since it is close to the spot sizes at airmass $m = 5$ (Figure 2).

## Appendix D: Symbols and Acronyms

C11        Celestron C11 Schmidt-Cassegrain telescope
M703       Alter M703 Maksutov-Cassegrain telescope made by Intes Micro
SPST05     starphotometer serial, fifth instrument built by Dr. Schulz & Partner GmbH
SPST06     starphotometer serial, same version as SPST05
SPST09     starphotometer serial, upgraded version
$D$        telescope diameter ("aperture" in Table 1)
$f$        telescope focal length
$f_\#$     telescope focal number (focal ratio)
$P_s$      telescope plate scale
FOV        telescope/instrument field of view
FWHM       Full Width at Half Maximum
px         pixel
bin        several pixels read together
$''$       arc seconds





|  |  |  |
|---|---|---|
| | $'$ | arc minute |
| | $e^-$ | electron |
| | $e$ | Euler's natural number |
| 1080 | ADU | analog-digital unit |
| | ASL | above sea level |
| | PSF | Point Spread Function |
| | $\omega$ | FWHM of the turbulence contribution to the long exposure star spot |
| | $\omega_s$ | FWHM of the turbulence contribution to the short exposure star spot |
| 1085 | $\omega_d$ | FWHM of the diffraction contribution to the star spot |
| | $r_0$ | length parameter of the turbulence |
| | $\lambda$ | wavelength |
| | $k_c$ | conversion factor radians to arc-seconds |
| | $k_t$ | turbulence parametrization constant |
| 1090 | $v$ | wind speed |
| | n | refraction index |
| | $k$ | absorption index |
| | $d$n | atmospheric layer contribution to n |
| | cnt | counts, instrument measurement unit |
| 1095 | $\theta$ | zenith angle |
| | $I_0, I$ | extra-atmospheric and attenuated star irradiances in absolute units |
| | $M_0, M$ | absolute extra-atmospheric and attenuated magnitudes |
| | $F_0, F$ | extra-atmospheric and attenuated star flux measurement |
| | $S_0, S$ | instrumental extra-atmospheric and attenuated magnitudes |
| 1100 | $I_{0,c}, I_{0,s}$ | exoatmospheric absolute irradiance of the catalog & instrument references |
| | $F_{0,c}, F_{0,s}$ | exoatmospheric instrument measurement of the catalog & instrument references |
| | $M_{0,c}, M_{0,s}$ | exoatmospheric catalog magnitude of the catalog & instrument references |
| | $S_{0,c}, S_{0,s}$ | exoatmospheric instrument magnitude of the catalog & instrument references |
| | $m$ | airmass |
| 1105 | $x$ | $m/0.921$ |
| | $\tau$ | atmospheric optical depth |
| | OD | atmospheric optical depth |
| | OSM | One-Star Method of observation |
| | TSM | Two-Star Method of observation |
| 1110 | $c$ | instrument specific photometric conversion factor |
| | $c_r$ | ratio of photometric system references |
| | $C$ | instrument specific photometric calibration parameter |
| | $\epsilon$ | observation error |



| | | |
|---|---|---|
| | $\delta_\epsilon, \delta_\tau$ | systematic observation and $\tau$ errors |
| 1115 | $\delta_{M_0}, \delta_x$ | systematic errors on $M_0$ and $x$ |
| | $\delta_S, \delta_C$ | systematic errors on $S$ and $C$ |
| | $\delta_t$ | systematic error on time |
| | $R$ | instrument measurement of the star irradiance |
| | $B$ | instrument measurement of the sky background |
| 1120 | $\delta_F$ | systematic errors on $F$ |
| | $\delta_R, \delta_B$ | systematic errors on $R$ and $B$ |
| | $g$ | ADU conversion factor |
| | $h$ | observatory altitude ASL |
| | $t$ | duration of an exposure |
| 1125 | $\delta_c$ | star centering tolerance |
| | $V$ | $M_0$ over the standard (visual) $V$ filter |
| | $U, B, J$ | as $V$, but for $U, B, J$ filters |
| | px | pixel |
| | ZP | zero-point of photometric system |
| 1130 | $z, z_t$ | apparent and true zenith angles |
| | $P$ | scattering phase function |
| | $\Omega$ | FOV/2 |
| | $\Delta\Omega$ | solid FOV angle |
| | $P\Delta\Omega$ | normalised $P$ over $\Delta\Omega$ |
| 1135 | $\overline{\omega}$ | Single Scattering Albedo |
| | SSA | Single Scattering Albedo |
| | IBC1–4 | aurora brightness classes |
| | R | Rayleigh unit |
| | $s$ | star spot off-axis or off-focus position |
| 1140 | $F_e$ | expected star signal when measured at standard stabilised temperature |
| | AERONET | AErosol RObotic NETwork of Cimel sunphotometers |
| | WMO | World Meteorological Organization |
| | Sp | spectral class of stars |
| | A,B,F,G | spectral class of stars |
| 1145 | Lum | luminosity class of stars |
| | HR | Harvard Revised Photometry star catalog |
| | HIP | Hipparcos star catalog |
| | Hp | Magnitude in Hipparcos photometric system |
| | RA(2000) | Right Ascension at epoch 2000 |
| 1150 | DEC(2000) | DEClination at epoch 2000 |





| | |
|---|---|
| $\sigma$ | standard deviation |
| $\theta, \theta_L$ | one-axis star spot jitter, low frequency jitter |
| $\sigma_{\theta_L}$ | standard deviation of low frequency jitter |
| $\omega_{\theta_L}$ | FWHM of low frequency jitter |
| 1155 $\sigma_\theta$ | one-axis standard deviation of the total jitter |
| $rad$ | radians |
| DU | Dobson Unit |
| TEC | Thermo-Electric Cooler |

*Author contributions.* This scientific investigation (conceptualization, methodology, formal analysis and publication) is essentially the work
1160 of L. Ivănescu (LI) with continuous oversight and insightful feedback from N. T. O'Neill (NO) and J.-P. Blanchet (JB). NO performed several
iterations of comprehensive revision on the paper. K.-H. Schulz (KS) built the starphotometer instruments and participated, with LI, in the
Eureka SPST09/C11 commissioning. The implementation, testing and development of a remote operation system as well as the mentoring
of the second generation SPST09/C11 instrument at Eureka and notably the amassing of a large, high-duty-cycle starphotometer database at
that challenging site was largely led by LI. K. Baibakov led the work on the first generation instruments at Sherbrooke, Egbert and Eureka
1165 and was a key player in the formulation of improvements needed to the Arctic-based starphotometers, their mount and dome.

*Competing interests.* No competing interests are present.

*Acknowledgements.* This work was supported, logistically and financially, by CANDAC (the Canadian Network for the Detection of Atmo-
spheric Change) through PAHA (Probing the Atmosphere of the High Arctic) funding, the NSERC CREATE Training Program in Arctic
Atmospheric Science and the Canadian Space Agency's (CSA's) Earth System Science - Data analysis (ESS-DA) program. We gratefully ac-
1170 knowledge the considerable instrumental and infrastructure assistance obtained from Pierre Fogal, Jim Drummond and the Eureka operations
staff.



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
