# Peer review of "Accuracy in starphotometry"

_Atmospheric Measurement Techniques, 2021_

## Community Comment (CC1)

Dear authors!

After reading of your interesting manuscript, I have some important remarks.

**1. The accuracy of Pulkovo's catalog**

1) At first. For meteorological purposes you must know only the **instrumental** extra-atmospheric stellar magnitudes for stars using for one-star daily monitoring. The $\triangle$-method usually is used for its obtaining on the first step (see point 3). In this case we must know only magnitudes' **difference** for two selected stars (from Pulkovo's catalog in our case).Therefore, only **inside** accuracy and homogeneity of catalog are important. But in your manuscript you analyze sufficiently correctly the **outside** accuracy of different catalogs. It is true, but it is not connected with our meteorological purposes. In description of our catalog we wrote:

"The extra-atmospheric spectral fluxes were transformed to the absolute scale. To covert the extra-atmospheric quasi-monochromatic fluxes in the instrumental system to absolute energy units, the basic catalog of spectrophotometric standards was used. This catalog consist of secondary and tertiary spectrophotometric standards. Vega ($\alpha$ Lyr, HR 7001, HD 172167) has been used as the primary standard. Although some authors have suspected this star to be variable, the latest observations of high accuracy at the Vilnius Observatory do not find any changes of its brightness (J. Sperauskas, personal communication).

        Several absolute calibrations of Vega are available (Kharitonov et al. 1968, Oke & Schild 1970, Tug et al. 1977, Hayes 1985). Some years ago, we performed our own absolute calibration of Vega with respect to National Primary Energy Standard of the USSR (Arkharov 1989; see Table 7). However, to preserve the uniform absolute system for all our seasonal catalogs, we always used the same energy distribution for Vega based on the absolute calibrations of Oke & Schild (1970) and Kharitonov et al. (1968). This compilative energy distribution is given in Table 1. The absolute energy distributions for the secondary standards were obtained by a direct tying up these stars to Vega and for the tertiary standards by tying up to some secondary standards."

From this text you can understand that data in Table 1 was used only as the same **constants** for convertation data from all seasonal catalogs to homogeneous photometric system, and its accuracy is not connected with inside accuracy of our catalog.

2) The accuracy for different stars in our catalog depends from number of observations. The list for monitoring must be select very careful. Usually we help to select best stars for any observational point. The best candidates are first, secondary, tertiary, extinctional standards, and non-variable stars with big number of observations (for example star HR 15 is not the best – it is known variable with period P=0.966222 d, 2.02-2.06 V). If the list of stars is selected correct, the accuracy will be sufficiently high.

**Due to mentioned above I can not agree with your conclusion that:**

**"The bright star catalog of extraterrestrial references is noted as a major source of errors with an attendant recommendation that its accuracy, as well as its spectral photometric variability, be significantly improved".**

**2. Bouguer method**

(according to http://www.astronet.ru/db/msg/1169494/node45.html#BougerErr), (in Russian)

If one and the same star is observed in monochromatic light with a wavelength $\lambda$ at two times $t_1$ and $t_2$, accordingly, at air masses $M(z_1)$ and $M(z_2)$, then the difference in the observed magnitudes, referred to the difference of the corresponding air masses, will give us the Bouguer coefficient of atmospheric extinction

$$\alpha(\lambda) = \frac{m(\lambda, t_1) - m(\lambda, t_2)}{M(z_1) - M(z_2)} \tag{1}$$

To obtain this coefficient, it is not necessary to know anything about the extra-atmospheric magnitude of the star, but if this magnitude is known (that is, the star is the standard), then

$$m(\lambda) - m_\circ(\lambda) = \alpha(\lambda)M(z). \tag{2}$$

Here is the equation of a straight line. This is the so-called *Bouguer line* . When a star with its motion passes various air masses on the celestial sphere, then the dependence of the attenuation $\Delta m$ in the atmosphere (in stellar magnitudes) from $M(z)$ is a straight line with a slope $\alpha(\lambda)$. Extrapolating this straight line to the value, $M(z) = 1$ we obtain the magnitude of the star at the zenith, and, continuing the straight line even further to the value $M(z) = 0$, we obtain its extra-atmospheric magnitude.

[Figure]

**Fig. 1**. Diagram of the Bouguer method and its errors when changing transparency

If atmospheric extinction were always constant in time, then the method would be perfectly ideal. But when the atmospheric absorption changes, the method can lead to gross errors. Let us assume for definiteness that during the observation session the extinction decreases. This is a fairly common phenomenon. From evening to late night, the temperature usually drops, droplets of water hanging in the atmosphere freeze and settle, daytime dust raised by human activity settles, etc. As the extinction coefficient decreases, the slope of the Bouguer lines decreases. Since the extra-atmospheric magnitude of the star does not depend on phenomena in the atmosphere, all these lines should converge at one point with coordinates [$M(z) = 0$; $\Delta m = 0$]. Since at the second, third and subsequent moments of time the slope of the Bouguer lines is different, the points can practically be on a strong straight line, but on another, and its extrapolation to $M(z) = 0$ will give an incorrect extra-atmospheric value.

The main disadvantage of the classical Bouguer method is that it does not work with varying extinction. Extinction changes are typical. At lowland observatories (in Pulkovo, or at the Lindenberg observatory) nights with stable transparency are rare. They do not always appear at high-altitude observatories too. For example, at the high-altitude Tien Shan observatory, it happens that a star that has already passed through the meridian continues to become brighter. Under these conditions, the best thing that can be done by the Bouguer method is to check whether there was a change in extinction or not. To do this, a star is observed, by which the extinction is determined, first to the meridian (here the star rises, the air mass decreases, the points creep up the Bouguer line), and then after (here the star sets, the air mass increases). If the extinction has not changed, the points will go back along the same straight line. Unfortunately, this is rarely the case. On opposite sides of the meridian, the points fall on straight lines with different slopes, which characterizes the change in extinction. Of these two slopes, the average is sometimes taken or the hypothesis is accepted that the change in the extinction coefficient occurs linearly with time, or according to some other law. But, as a rule, the extinction changes quite arbitrarily, and these tricks still do not lead to an increase in accuracy.

**3. $\Delta$ - method (Pulkovo's version) and its using for meteorology.**

Let's try to get rid of the influence of the varying extinction. Obviously, one star will not be enough. Firstly, the star will rise for a very long time, then it will go down for a long time. Naturally, the idea arises to observe practically at the same time instant (quickly one after the other) two stars at different zenith distances. This method is also called the $\Delta$-method, the high and low star method, etc.

Let there are two stars A and B, when the values of extra-atmospheric magnitudes for its are known (from Pulkovo's catalog in our case). So, one can have from one pair-dimension of stars A and B:

$$\alpha(\lambda) = \frac{[m_A(\lambda) - m_B(\lambda)] - [m_A^\circ(\lambda) - m_B^\circ(\lambda)]}{M(z_A) - M(z_B)}. \tag{3}$$

It is advantageous to measure one standard near the zenith, and the second (usually in several minutes) rather low above the horizon. You should not choose the second star too low, as other distortions of the luminous flux will interfere there. It is convenient to choose a low star at an air mass no more 4. Observations of two stars are carried out quasi-simultaneously, and thus, using

the pair method, we can determine the *instantaneous* value (for $t_{AB} = (t_A+t_B)/2$ i.e. $\alpha_\lambda(t_{AB})$) of the atmospheric extinction coefficient. I. e., we suppose that atmospheric extinction is the same for all direction during one pair-dimension only. **This method must not be used for daily atmospheric monitoring.** We use this method only for definition of the first approximation of values $\alpha_\lambda(t_{AB})$ during nights selected for obtaining **instrumental** extra-atmospheric magnitudes **m₀** (needed for one-star monitoring). On the first step we obtain the polynomial dependence $\alpha_\lambda(t_{AB})$ for all night. By analysis of this dependence we can select the nights with slow extinctional changings. Then we calculate for all observations during selected night (by using this polynomial dependence) the individual values $\alpha_\lambda(t_A)$ , $\alpha_\lambda(t_B)$ ,and so on.

It is important to note, that in this case we use catalog data for calculation the value of magnitudes' differences ONLY. For monochromatic (ore quasi-monochromatic in our case) light these differences will be the same for all photometric systems. So, we can use the modification of formula (2) to calculations of **m₀** for every star used:

$$m_0(\lambda) = m(\lambda) - \alpha(\lambda) M(z) \tag{4}$$

Using (4) and individual values $\alpha_\lambda(t_A)$ , $\alpha_\lambda(t_B)$ ,and so on, we can calculate individual values **m₀** for all observations of every star during night. Then we calculate mean values $\overline{m_0}$ for every star as first approximation for **instrumental** extra-atmospheric magnitude.

And as the last step of first approximation we calculate new individual values $\alpha_\lambda(t_i)$ using $\overline{m_0}$ as **m₀** and formula:

$$\alpha(\lambda) = \frac{m(\lambda) - m_0(\lambda)}{M(z)} \tag{5}$$

These results we can use as first step for second approximation (iteration).

For most nights the three iterations are sufficient for obtaining good mean values $\overline{m_0}$ for all stars used.

**4. Two small corrections.**

1) Line 12 "i.e. at airmasses lower than 5" is not corrected. Must be "i.e. at airmasses less than 5"

2) Line 156 "($V < 3$)". More correct will be "($V < 6$)".

I very hope that my remarks will be useful for you, and will take its into account during correction of final text of your manuscript.

With best regards,

Victor Novikov.

Pulkovo observatory.

novikov_victor@mail.ru

---

## Author Response (AR1)

**Relevant changes made to the manuscript**

The relevant changes made to the paper are listed bellow. For page identification and performed changes, please refer to the attached "track-changes document" (our filename: accuracydiff.pdf), where the strikethrough red text means removed text, and the blue text means added text. To facilitate the addition of cross-reference hyperlinks to the concerned referee comments (RC1, RC2 and CC1), a color-coded version of the published responses are following on the next pages.

- Page 1: abstract wording change to address CC1 concerns
- Page 2-3: changes to address RC2 and CC1 concerns
- Page 6: changes to underscore that telescope optical aberrations were supposed to be negligible with respect to atmospheric turbulence effects, but not actually measured
- Pages 11-13: Section 3.4 (Measuring methods) was significantly upgraded to add new information from CC1 and the references it mentioned. This also addresses RC1 comments.
- Pages 15-20: Section 4 (Spectrophotometric catalog ($M_0$) accuracy) was divided in subsections to facilitate concept identification – necessary to address comments from CC1 and RC2.
- Pages 24-26: Section 6.3 (Forward scattering) was updated, mainly with a footnote in page 24 (to address RC2 concerns) and a last paragraph (to address RC1 concerns).
- Page 29: Section 7.1 title "Misalignment issues" was changed to "Starlight vignetting", that is more specific for our type of instrument.
- Page 39: Channel 17 updated in the table to address RC2 concerns
- Page 41: Channel 17 updated in the list to address RC2 concerns
- Pages 41-42: Introduced a table summarizing all the sources of errors, as requested by RC2
- Page 43: Added OD of $NO_2$ at 500 nm, as requested by RC2
- Page 44: Conclusions was updated to the reflect the changes
- Page 45: Change requested by RC2
- Pages 51-56: Appendix D (Symbols and Acronyms) was split by Symbols and Acronyms, and rearranged in alphabetical order, as requested by RC2
- Pages 58-66: New references were added corresponding to the paper changes

Response to Referee Comment #1 (RC1)

The referee's comments are presented in black and our answers are written in blue. Modifications of the manuscript are shown in red.

In this paper "Accuracy in starphotometry", the authors conducted a comprehensive and thorough study of error sources that affect optical depth (OD) retrievals using starphotometers, and further recommended favorable observing conditions including identifying 20 channels for mediating some of the error sources and improving accuracy in OD retrievals using starphotometers. I am not an expert in starphotometry; thus, I focus my comments on OD retrievals in general and hope other reviewers can comments on starphotometry related discussions. But in general, this is a well written paper that highlights various sources of error in starphotometry. The content of the paper is a significant contribution for further improving accuracies in starphotometery. I recommend publication of the paper after some minor corrections.

We thank the referee for the careful reading and the generally positive review!

Comments:

- Thin cirrus cloud contamination can be a problem for sun-photometer data (Chew, B. N., Campbell, J. Reid, D. M. Giles, J. Welton, S. V. Salinas, and S. C. Liew (2011), Tropical cirrus cloud contamination in sun photometer data, *Atmos. Environ., 45*, 6724-6731, doi:10.1016/j.atmosenv.2011.08.017). Is this thin cirrus cloud contamination also a problem for starphotometry? Based on the paper, it seems both cloud and aerosol OD can be derived. How, then, do the authors perform the scene identification? Are there error sources related to misidentification of thin clouds and aerosols?

Thin cirrus may, from the perspective of our paper, induce two notable effects:
- an increase in coarse mode (CM) AOD with no change in the fine mode (FM) AOD. Putting aside, for the moment, the forward scattering error, there is no FM feature misidentification if one employs the (AERONET) SDA technique to back out the FM AOD. In cloudy scenes, one expects that cloud particles dominate the CM OD component. There is always some ice crystal presence in the air during the polar winter: this will likely dominate any CM aerosol presence (c.f., for example, O'Neill et al., 2016). We never explicitly observed large CM polar winter AODs: CM AODs are typically small (sub 0.01) and dominated by local dust and/or sea-salt during the spring and fall (Aboel-Fetouh et al., 2020). Put simply, we do not try to measure CM AOD during the polar winter without complementary CM data retrieved from, for example,

lidar profiles (c.f. Baibakov et al., 2015), surface PSD (particle size distribution) measurements  or (in the case of sunphotometry) almucantar scans.

- A "forward scattering error" (see the "Forward scattering" section of our submitted manuscript). The very small starphotometer FOV largely ensures that this kind of error is generally negligible.

In the "Forward scattering" section of the revised manuscript we added some text to clarify the concept of the associated error (notably that it was only a problem in the presence of large-size cloud crystals and that accounting for multiple scattering contributions to that type of error were not a problem in starphotometry). A new paragraph was also inserted at the end of that section to clarify that the SDA algorithm could be employed to separate FM AOD from CM OD and that very-small FOV starphotometry was uniquely positioned to extract FM AOD (but not CM AOD) even in the face of the forward scattering error.

- In section 8, the authors discussed optimal channel selections and provided recommendations for achieving OD accuracy of 0.01. Are the recommendations the same for the TSM and OSM methods?

TSM provides an intermediate OD measurement for two observation directions. While the recommendations generally concern both methods, TSM is typically less accurate. To emphasize the OSM *vs* TSM accuracy difference (and to address concerns from other referees), the "Measuring methods" section 3.4 was significantly updated.

- Eventually, either aerosol or cloud OD will be derived. This requires an understanding of Rayleigh OD, which is also a function of observing conditions. How much is the error in Rayleigh OD calculations based on the available observations associated with starphotometers?

For our spectral range, the largest Rayleigh OD of 0.37 occurs at 400 nm. A surface pressure measurement error of 30 mb is required in order to limit the Rayleigh OD error to 0.01 (Bucholtz, 1995). Since such measurements are performed with 1 mb accuracy at Eureka, the associated starphotometer retrieval errors are expected to be negligible. Even if there are one-hour gaps between pressure measurements, the interpolation errors are generally within the same error margins. In the rare case of a low-pressure front crossing the site, Rayleigh errors could become significant (but the sky will likely be too cloudy to perform starphotometer measurements). Since this error can be neglected, we did not make any modifications in the revised article.

**References**

Please see the Reference section of our paper for citations that are not in the list below!

AboEl Fetouh, Y., N. T. O'Neill, K. Ranjbar, S. Hesaraki, I. Abboud, V. Fioletov, P. S. Sobolewski, Climatological-scale analysis of intensive and semi-intensive aerosol parameters derived from AERONET Arctic retrievals, JGR, 125(10), p.e2019JD031569, 2020.

O'Neill, N. T., K. Baibakov, S. Hesaraki, L. Ivanescu, R. V. Martin, C. Perro, J. P. Chaubey, A. Herber, and T. J. Duck. "Temporal and spectral cloud screening of polar winter aerosol optical depth (AOD): impact of homogeneous and inhomogeneous clouds and crystal layers on climatological-scale AODs." ACP, 16, no. 19, 12753-12765, 2016.

**Response to Referee Comment #2 (RC2)**

The referee's comments are presented in black and our answers are written in blue. Modifications of the manuscript, if any, are reported in red.

In this paper "Accuracy in starphotometry", the authors present a detailed and comprehensive study of error sources for retrievals of the optical depth (OD) using the starphotometer technique. Based on this advanced quantification of errors impacts, the authors give some recommendations regarding maintenance, conditions of utilisation, calibration, observation techniques in order to reduce the uncertainties. A spectral aspect is discussed, that is very important for the starphotometry community: the pertinence of the existing catalogs of star magnitudes for the use of starphotometers, possible improvements and how to deal with all the discussed difficulties (choice of the resolution of the catalog; choice of the spectral channels that allow accurate inversions of the OD).

Despite some minor and very specific suggestions for improvements that I will explain in my comments, this is a well written paper, both considering the scientific quality (analyses, equations) and considering the quality of the English and the clarity of the text. Thus, I consider that this paper is an important contribution for enhancements in the use and for the accuracy of photometry techniques for OD retrievals. I recommend publication of this paper after some minor corrections.

We thank the referee for the in-depth reading, as well as for the insightful and generally positive feedback!

Comments:

- Observational error level of 1%

In the abstract (Line 2), since the beginning of the introduction (Line 24 and after in Line 47) and during the whole article, you set the goal of the accuracy of this technique in "observational error level of 1%: a spectral optical depth (OD) error level of 0.01 level of". I have two comments/questions about that:

1) Please define what is the "OD": Is it "AOD" (Aerosol Optical Depth) or "COD" (Cloud Optical Depth) depending on what you want to retrieve, or is it the optical depth like considering the optical path interpretation (OD = ln(I/I0)), or is it the "TOD" (total optical depth: columnar optical depth): TOD = AOD + COD + tau_rayleigh + tau_gas + ... = ln(I/I0)/airmass?

"τ (total vertical optical depth") was defined on page 8, Line 169 of the submitted manuscript. We used τ and OD as synonyms (same definition in the Symbols and Acronyms list in Appendix D). In order to clarify the slant-path versus columnar ambiguity, τ and OD were explicitly defined as "vertical (columnar) optical depth" in the Symbols and Acronyms list, as well as in the Introduction. Text (with footnotes) was also added to the Introduction to explicitly define (i) the scattering and absorption (extinction) components of any optical depth and (ii) our speciated optical depth acronyms.

2) Explain briefly in introduction why you want a value of 0.01 as goal of this "observational error level". I suggest to look at WMO recommendation about the error on AOD (Aerosol Optical Depth), depending on the airmass (m): Delta_AOD must be < 0,005 +/- 0,001m (Formula can be found in Kazadzis, S., Kouremeti, et al. 2018, Results from the Fourth WMO Filter Radiometer Comparison for aerosol optical depth measurements. Atmos. Chem. Phys. (5), 3185–3201). From this formula of recommendation on AOD error, you can find out the most strict airmass condition, and compute the acceptable error on the OD that result of it.

The reasons behind our 0.01 goal were explained in Lines 23-25, page 1-2 of the submitted manuscript: we wished to limit the accuracy error to the 0.01 precision error inferred from Figure 4 of O'Neill et al., (2001). This is consistent with the WMO criteria for a high star with a typical airmass value of m=2 (inserted in the WMO δτ expression of 0.005 + 0.01/m). It's also consistent with the satellite AOD retrieval requirements for climate energy budget analysis (Chylek, 2003). In order to address this concern, we added a sentence detailing the last two 0.01 constraints immediately after the O'Neill et al. (2001) sentence in the Introduction.

- About "C"

You introduce the parameter C ("instrument specific calibration parameter") in Line 193 (in 3.3. Practical considerations). This is maybe the most important parameter for operational retrieval with a starphotometer. During the whole article, you assume that C is not star dependent: you use the same C for the two different stars in the TSM method for instance. This assumption (C is the same for two different stars) may be acceptable under some conditions that are mainly respected in the star photometry. One condition is that the channels are relatively narrow so that the convolution of the instrumental response function with the spectrum of the star magnitude is the same for the two stars that have different spectra of star magnitude. I think it is worth to give an information about below which value of bandwidth the assumption is valuable;

cherry on the cake would be a quantification of the possible error that can result for a larger band or for different convolution of response function with star spectra (in case of big differences of star spectra inside the spectral band of the channel). This assumption should be remembered when you explain the basics of the TSM in equations (25) and (26) (Line 258 and 261, at the end of the paragraph 3.4.2). Again, you write this assumption without proof or discussion at line 286 (Beginning of Part 4): "the more convenient star-independent calibration in terms of C".

Errors related to bandwidth size were indeed found to be negligible in section 6.1 (Heterochromaticity) of the submitted manuscript. The errors related to the convolution of the instrumental response function with the star spectrum were presented in section 4 (Spectrophotometric catalog ($M_0$) accuracy). This response also addresses the reviewers "the more convenient star-independent calibration in terms of C" comment. Section 4 of the revised manuscript was divided into four subsections in order to better underscore its key elements[1].

We consider that the spectrophotometric catalog errors (including the mismatch error) are a major limitation to improving the OD measurement accuracy. For this reason, as the referee also noted, C cannot be simply retrieved from equation (25), at least not from a single pair of stars. Significant revisions were made to Subsection 3.4.2 (TSM) in order to address the issues raised above (particularly in terms of better detailing the different options available for retrieving accurate C and $\tau$ values in the face of $M_0$ errors).

- Forward scattering error

Question about Figure 14 and the discussion about it at the end of paragraph 6.3: you consider delta_tau/tau as the important parameter and you look [at] the forward parameter part. Is it only a formula that is plotted on the figure, or are there the results of a real irradiances computation with a radiative transfer code? A proper radiative transfer simulation would have the benefit to consider not only single scattering, but also multi-scattering and scattering between the different layers.

We thank the referee to point out that this aspect is worth mentioning it. Figure 14 is based on equation (36): a purely single scattering result, which is an entirely appropriate approximation for the cirrus type crystals, which can significantly decrease their measured OD. As we point out in a footnote of the revised manuscript, multiple scattering plays no significant role in the forward scattering error in the case of starphotometers.
* * *
[1] with "Bandwidth mismatch error" being one of those subsections

- Table summarizing all sources of errors

Before 8.2 (recommendations): Here it would be welcome to have a table that summarizes all sources of errors that have been quantified above, with the values of the possible errors considering different way of dealing with the instrument (calibration often or rare, weather conditions, elevation of the stars, etc...).

We thank the referee for this good idea! Such a table was created and inserted at the beginning of the Starphotometry recommendations section.

- Appendix D: Symbols and acronyms

Please make two tables: one for the symbols used in equations (tau, omega, f, etc...), and one for the acronyms, and please sort both of them in alphabetical order!

We thank the referee for this good idea! It was implemented accordingly.

- Minor comments/typos

Line 28: "Sunphotometry, and to some extend moonphotometry, are much more mature technology" -> Moonphotometry (after 2013) is less mature than starphotometry (beginning of the 90ies)

While having its particular issues and challenges, moonphotometry is much closely related to sunphotometry, and is able to inherit several of its advantages. In order to address this issue, we replaced that sentence with "Sunphotometry, and to a certain extent moonphotometry, are much more mature technologies".

Line 298: Typo: "shorcomings" -> *shortcomings

It was corrected accordingly.

Line 584: Problems are mentioned above 1000 nm, what is not a big issue considering the range of the SPST starphotometers

The SPST starphotometers have the ability to go beyond 1100 nm (see Figure 26), but the sensitivity is very low and this may only be possible with a cold star such as Procyon. However, one can work with most stars up to about 1050 nm. The range beyond 1000 nm is useful for anchoring coarse mode OD calculations (especially in the case of cloud particles) and to make a better base-line estimations for water vapor retrieval.

Line 774: You give the value of tau_NO2 for 400 nm, please give also the value at 500nm, since the order of magnitude of this parameter is better known at this wavelength (standard of the community)

Based on the cross-section spectrum of Burrows (1998), the 500 nm $\tau_{NO2}$ value is a factor of 3 smaller than the value at 400 nm. A summertime 500 nm $\tau_{NO2}$ value at Eureka would then be 0.001 while a wintertime value is expected to be even smaller.

However, at low latitudes, it may be as high as $\tau_{NO2}$=0.01 at 500 nm. In order to address this concern, in the 8.2 Starphotometry recommendations subsection, we replaced "(i.e. up to 0.03 OD at 400 nm)" by "(i.e. up to 0.03 OD at 400 nm, or 0.01 OD at 500 nm)".

Table 2, Channel 15: "almost WMO lambda" is truer than "WMO lambda" (20 nm shift)
Corrected accordingly.

Table 12, Channel 17: 936 nm is also an AERONET standard (935 nm is used by AERONET, but only for the PWV retrieval, not for the AOD, thus if you want to compare starphotometer and AERONET for WV, this channel is the most important one)
Addressed accordingly (in the table and in the corresponding description list).

Line 947 (Appendix A1): "at the Lindenberg observatory in Germany" -> *at the Deutscher Wetterdienst (DWD) Meteorological Observatory of Lindenberg (Germany)
Text was corrected.

Line 1065 (Appendix D: Acronyms): SPST = Schulz and Partner STarphotometer (or "Schulz and Partner STernphotometer" in German)
Text was added.

Line 1072 (Appendix D: Acronyms): FOV = "Field Of View"
It was modified accordingly.

References

All references not found below can be found in the references section of the revised paper.

Burrows, J.P., A. Dehn, B. Deters, S. Himmelmann, A. Richter, S. Voigt, J. Orphal Atmospheric remote-sensing reference data from GOME: Part 1. Temperature dependent absorption cross-sections of NO2 in the 231–794 nm range, J. Quant. Spectrosc. Rad. Transfer, 60, pp. 1025-1031, 1998.

**Response to Community Comment #1 (CC1)**

The referee's comments are presented in black and our answers are written in blue. Modifications of the manuscript, if any, are written in red.

We thank the reviewer for the careful reading as well as the unique descriptions and references: this helped to better articulate our reactions to his or her comments!

Comments:

1. The accuracy of Pulkovo's catalog

In order to better underscore the significance of this subject in our paper, we added an explicit subsection title 4.1 Pulkovo catalog errors.

> 1) Outside accuracy
>
> The "outside accuracy" that we identified as a catalog bias will not, we agree, actually affect the optical depth measurements (see lines 312-313 in page 13 of the submitted manuscript).
>
> 2) Inside accuracy
>
> The "inside accuracy" that we interpret as the Figure 4 standard deviation of ~0.02[2] is in agreement with the Pulkovo catalog findings of Alekseeva et al. (1996)[3]. This value, the reviewer will appreciate, falls short of our stated goal of 0.01 OD accuracy. We recognize that the Pulkovo catalog probably represents the most accurate bright star catalog available. Nonetheless, improvements (specifically the identification of stable stars whose OD uncertainty is significantly smaller than 0.02) are required.

2. Bouguer method

We thank the reviewer for the Mironov (2008) reference with its important synthesis of several calibration methods.

We incorporated that citation, as well as the Gutierrez-Moreno and Stock (1966) and Stock (1969) citations, in our Introduction and the 3.4.2. TSM subsection.

3. Δ - method (Pulkovo's version) and its using for meteorology

We thank the reviewer for the manuscript on the unpublished[4] Pulkovo iterative calibration method. The method detailed the retrieval of extra-atmospheric instrumental magnitudes. This provides a set of magnitudes that would enable the creation of a new extra-atmospheric star catalog. Such a task should give better results at a high-altitude observatory. On the other hand, if a catalog is already accurately
* * *
[2] represented by the blue and red shading of Fig. 4 (whose amplitude is a bit higher in the NIR).

[3] see line 316 on page 14 of the submitted manuscript

[4] unpublished, as far as we know of.

predetermined, only the retrieval of single, star independent, calibration parameter (C), would be required. This task may be feasible even at a low-altitude observatory.

In order to better articulate our intention to use a star independent calibration, and to incorporate the reviewer's concerns (and those of the other reviewers), we made a significant update of the 3.4.2 TSM subsection, which includes a reference to the Pulkovo method.

4. Two small corrections
   1) Line 12 "i.e. at airmasses lower than 5" is not corrected. Must be "i.e. at airmasses less than 5"

      It was corrected accordingly.
   2) Line 156 "($V < 3$)". More correct will be "(V <6)".

      Even if the Pulkovo catalog incorporated V<6 stars, our star dataset selection is limited to V<3. Arctic stars with V>2 are generally difficult to use as low stars due to our small telescope diameter (11 inches) and to the limitation of the current detector.

      No changes were accordingly applied to the paper.

References
Please see the Reference section of our paper!